# FACTS IN STATS: IMPACTS OF PRETRAINING DIVERSITY ON LANGUAGE MODEL GENERALIZATION

## ABSTRACT

Language models are pretrained on sequences that blend statistical regularities (structures making text fluent) with factual associations between specific tokens (corresponding to knowledge of facts). While recent work suggests that the variability of their interaction, such as paraphrases of factual associations, critically determines generalization ability, we lack a systematic analysis of these impacts. This paper introduces a flexible synthetic testbed that combines a statistical stream of generic tokens with an abstract factual stream of source-target token pairs, enabling fine-grained control over their interaction. Specifically, the design enables the independent control of diversity nature by manipulating stream composition (contextual structure) and the level of diversity by varying which statistical streams each fact appears in. Through controlled experiments, we find that while higher contextual diversity delays in-distribution (ID) factual accuracy, its effect on out-of-distribution (OOD) generalization depends critically on contextual structure. In some cases, OOD performance follows the same trend as ID, but in others, diversity becomes essential for non-trivial factual learning. Even when low diversity prohibits factual recall, optimal diversity levels depend on training duration. Beyond factual recall failures, we identify structures where statistical generalization fails independently, and others where both capabilities degrade simultaneously. This demonstrates how the interplay between contextual design and diversity level impacts different aspects of generalization. Furthermore, through a series of controlled interventions on the model components, we trace the generalization failures to distinct optimization bottlenecks, highlighting the importance of the learned embeddings and the unembedding layer. Overall, our synthetic framework allows us to isolate effects that would be confounded in large-scale studies, thus offering a controlled testbed for future investigations.

## 1 INTRODUCTION

Modern transformer-based language models (LMs), trained on massive corpora of natural-text sequences, simultaneously learn to generate contextually plausible sequences that follow statistical and linguistic patterns, while also learning significant amount of real-world knowledge. Their parameters effectively become an implicit knowledge base that can rival structured knowledge bases in question-answering and recall tasks (Petroni et al., 2019; Roberts et al., 2020; Dai et al., 2021; Allen-Zhu and Li, 2023; Akyürek et al., 2022; Yang et al., 2024; Mallen et al., 2023; Meng et al., 2023). This dual capability is remarkable precisely because the LM receives no explicit fact supervision during training: it never encounters knowledge graphs, labeled facts, or any explicit differentiation between linguistic and factual content. Yet somehow, it develops the ability to both follow linguistic patterns and encode world knowledge. *How does the conceptually-simple next-token prediction objective enable a language model to implicitly learn both linguistic structure and factual knowledge? Are there inherent trade-offs in acquiring these two types of information?*

Intuitively, the repetition of factual information across varying and diverse linguistic contexts during training, essentially appearing as *paraphrases* of the same underlying facts, likely plays a key role in facilitating factual recall from pure next-token prediction supervision. But, this raises several questions:

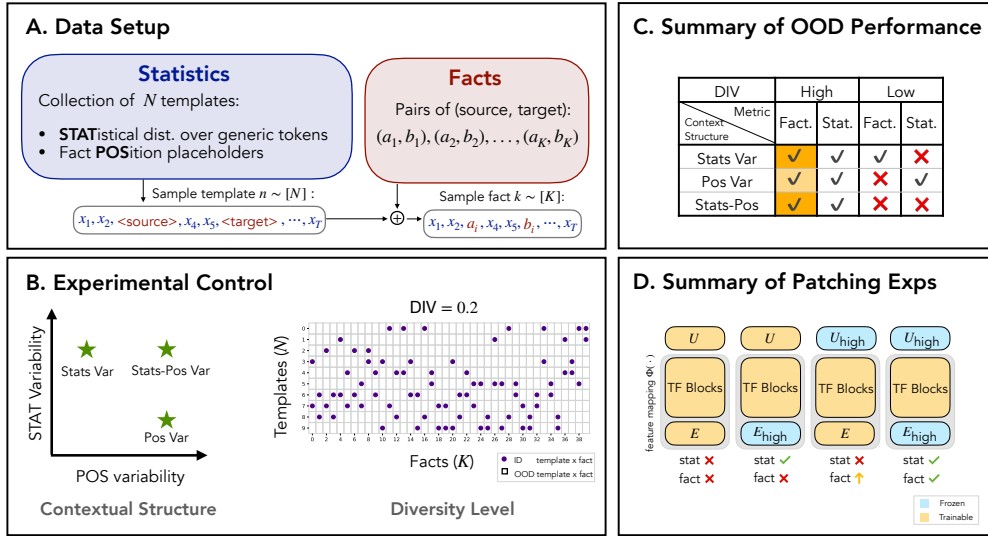

**Figure 1: Proposed testbed and summary of key findings.** **(A)** Sequences are generated by combining a **factual stream**, a set of atomic pairs, and a **statistical stream** of templates, each a statistical distribution over sequences with two fact placeholder positions. Final sequences insert a fact into a sequence sampled from a template (Sec. 2). **(B)** We control two key data properties. **(Left) Contextual structure:** The degree of statistical and (fact) positional variation across templates. We explore three structures: **Stats Var** (varied statistics, fixed positions), **Pos Var** (fixed statistics, varied positions), and **Stats-Pos Var** (varied on both axes). **(Right) Diversity level:** The number of unique templates each fact is paired with in the training corpus. The table shows an *exposure matrix*, where dots indicate template-fact pairs used for training (Sec. 2.1-2.2). **(C)** Low diversity impairs out-of-distribution generalization, but the specific aspects affected (factual recall vs. statistical pattern generation) depend on contextual structure. At high diversity, all structures enable both types of generalization, though with varying efficiency (colored boxes: darker means more training iterations needed with higher DIV) (Sec. 3). **(D)** While training on low-diversity data impairs statistical and factual generalization, when the embeddings or unembeddings from a high-diversity model are "patched" in, the remaining modules can still learn a generalizing solution even when trained on that same low-diversity data (Sec. 4).

*How frequently must a model encounter a fact during training to reliably learn it, and how does this depend on the* diversity *of linguistic contexts (paraphrases) in which the fact appears? Under what training conditions is it possible to isolate factual associations from linguistic structures? Moreover, is diversity only possibly impacting factual recall, or can it also impact the model's ability to learn statistical patterns?*

The intriguing factual-recall capability of LLMs has motivated several recent studies probing how models acquire, represent, and recall facts using controlled setups. Allen-Zhu and Li (2023) and very recently Zucchet et al. (2025) study synthetic biography corpora with fixed templates and varying factual associations (e.g., name, age, occupation) to examine how paraphrastic diversity affects factual recall. While their data setup provides more control than raw natural-text corpora, where linguistic patterns and facts are intricately interleaved, they still operate at considerable scale, requiring transformers with 8-12 layers/8-12 heads/512-768 dimension for empirical analysis. This limits accessibility and makes comprehensive parameter sweeps over controllable variables computationally challenging. Additionally, by employing fixed templates, these setups isolate focus on factual learning while ignoring the model's ability to simultaneously acquire linguistic structure and leaving fine-grained impacts of diversity on learning of both components underexplored.

Motivated by these, we develop a small-scale synthetic testbed for studying at a fine-grained level the interactions between statistical and factual learning streams, specifically as captured in terms of context *diversity*. Our framework offers three key advantages. *Tractability:* Its minimal scale enables comprehensive parameter sweeps and in-depth analysis that would be computationally challenging in large-scale setups. All our experiments use 4-layer/4-head/32-dim transformers, and phenomena can be reproduced even with 1-layer transformers. All experiments are performed on on single Tesla V100-SXM2-GPU(16GB) and can be replicated on the free-version of Google-Colab. *Reproducibility:* Despite its minimalism, our framework reproduces empirical phenomena from large-scale studies, including stage-wise factual learning (Zucchet et al., 2025) and improved factual recall with larger diversity (Allen-Zhu and Li, 2023). *Experimental control:* The explicit statistical stream enables

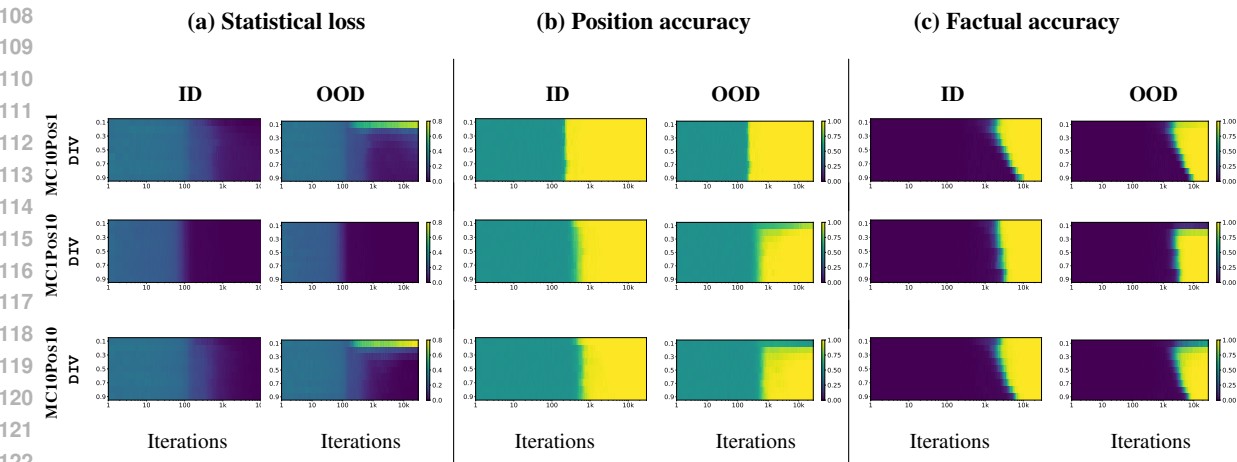

**Figure 2: Generalization performance for varying diversity levels and context structures.** Heatmaps show **(a)** $\mathtt{Loss}_{\mathrm{stat}}$, **(b)** $\mathtt{Acc}_{\mathrm{pos}}$ and **(c)** $\mathtt{Acc}_{\mathrm{fact}}$ over training iterations (horizontal axis) and diversity level (vertical axis) for the three contextual structures defined in Sec. 2.2: (top) MC10POS1 with only statistical variations, (middle) MC1POS10 with only positional variations and (bottom) MC10POS10 with both types of variations across templates. Each metric is displayed for in-distribution (ID, left column) and out-of-distribution (OOD, right column) fact–template pairs. Brighter colors denote higher *loss* for $\mathtt{Loss}_{\mathrm{stat}}$ and higher *accuracy* for $\mathtt{Acc}_{\mathrm{pos}}$ and $\mathtt{Acc}_{\mathrm{fact}}$. Increasing diversity can slow down convergence, whereas very low diversity leads to OOD failure on one or multiple metrics. See Sec. 2.3 for metrics and Sec. 3 for details.

exhaustive exploration of how pretraining data diversity, both in terms of exposure level and structural variety, impacts transformer out-of-distribution generalization under next-token prediction training.

In broader perspective, our methodology mirrors research turning to deliberately small-scale tasks and transformers to study other next-token prediction phenomena, such as in-context learning (Garg et al., 2022; Edelman et al., 2024b), grokking (Power et al., 2022; Nanda et al., 2023), and arithmetic reasoning (Lee et al., 2023). Within factual recall research, our data setup bears some abstract similarity to Nichani et al. (2024), who use synthetic tasks to analyze scaling laws and dynamics of factual recall. Yet, their setup lacks statistical learning, any account of diversity effects, and operates in non-autoregressive setting. See App. A for detailed comparison to related works.

### 1.1 SUMMARY OF METHODOLOGY AND FINDINGS

Our data framework factorizes sequences into a *statistical* stream and a *factual* stream. We model the statistical stream as a mixture of $N$ *template* distributions over token sequences. Each template generates a sequence from a template-dependent Markov Chain (MC) distribution and reserves two designated positions for fact insertion; these positions may vary across templates to capture different structural patterns. The factual stream consists of *source–target* token pairs drawn from distinct vocabularies that are inserted in a template's designated positions to generate complete sequences. This gives precise and independent control along two axes: (a) **Diversity level**—how many different templates each fact appears in during training, and (b) **Contextual structure**—how template formats vary in their statistical and positional properties. See Fig. 1-(A,B).

We train small transformers on this data and evaluate generalization through carefully designed prompts, testing three distinct aspects: **(1) Factual recall accuracy**—the model's ability to produce the correct fact target for a given source in the prompt; **(2) Position accuracy**—adherence to the specific positional patterns of the prompt template; and **(3) Statistical accuracy**— adherence to the statistical distribution of the prompt template. We evaluate performance on both **in-distribution (ID)** data, where fact-template combinations match training data, and **out-of-distribution (OOD)** data, where facts appear in novel templates. See Sec. 2 for framework details. We summarize our key findings (also see Fig. 1-(C,D)):

**Impact of diversity**. The diversity of the training data has several effects on generalization:

1. Higher diversity can slow ID factual recall while leaving ID statistical accuracy largely unaffected. (Fig. 2 and Sec. 3.1).

2. OOD effects depend on contextual structure: Low diversity can severely impact factual recall, statistical accuracy, or both. (Figs. 1-2, and Sec. 3.2).

3. Data diversity introduces a cost-benefit dynamic: Higher diversity guarantees better performance but only when provided with sufficient training budget. With limited training time, intermediate diversity levels are optimal. (Figs. 2-3 and Sec. 3.2).

4. The learning dynamics of the two streams (statistical and factual) are coupled: Increased complexity in one stream, delays learning of both streams. (Fig. 4 and Sec. 3.3).

**Optimization bottlenecks.** Architectural parameters that achieve perfect OOD generalization exist regardless of diversity level. However, diversity shapes the optimization path, determining whether and how quickly training reaches those parameters. Through targeted interventions, we localize the bottlenecks to specific architectural modules. (Figs. 6-5 and Sec. 4).

5. Training on low-diversity data produces poor last-layer features; retraining only the unembedding layer on high-diversity data fails to restore generalization.

6. Poor optimization of the attention and embedding modules degrades statistical and positional generalization.

7. The embedding and unembedding layers are jointly responsible for factual-recall failures; improving either in isolation fails to recover performance, but joint intervention on both helps with generalization.

8. The unembedding layer causes the slowdown in factual recall at high diversity; training features alone with fixed unembeddings shows no such slowdown.

## 2 SETUP

We consider sequences $\mathbf{x} = (x_1, ..., x_T)$ of length $T$, produced by mixing two independent sources: a *factual* and a *statistical* stream, with each token $x_t$ belonging to a global vocabulary $\mathcal{V} = [V]$.

**Factual stream**. We specify $K$ atomic facts $\mathcal{K} := \{(a_k, b_k)\}_{k \in [K]}$ each given as an ordered *source-target* pair of distinct tokens $a_k, b_k \in \mathcal{V}$. The vocabulary of facts is $\mathcal{V}_{\mathcal{K}} = \{a_k\}_{k \in [K]} \bigcup \{b_k\}_{k \in [K]}$.

**Statistical stream (Templates)**. Background sequences $\mathbf{z} = (z_1, \ldots, z_T)$ to which facts are later embedded are sampled from a mixture of $N$ template distributions: $\mathbf{z} \sim \mathcal{D} = \frac{1}{N} \sum_{n=1}^{N} \mathcal{D}_n$. Formally, a template $\mathcal{D}_n$ is a probability distribution over length-$T$ sequences drawn from the *generic vocabulary* $\mathcal{V}_{\mathcal{D}} := \mathcal{V} \setminus \mathcal{V}_{\mathcal{K}}$ of size $V_{\mathcal{D}} = |\mathcal{V}_{\mathcal{D}}|$, with two marked positions $\mathbf{I} = (i, j) \in [T] \times [T]$, $i \leq T/2 < j$ reserved for the source/target fact tokens. For concreteness, unless otherwise stated, we assume the generic tokens are produced by a first-order Markov chain (MC). Thus, a template $\mathcal{D}_n$ is defined by: **(i)** a transition matrix $\mathbf{P}_n \in \Delta^{V_{\mathcal{D}} \times V_{\mathcal{D}}}$, and **(ii)** a placeholder position pair $\mathbf{I}_n$. We sample from $\mathcal{D}_n$ by generating a length $T-2$ MC sequence from $\mathbf{P}_n$, and reserving the positions $\mathbf{I}_n$ for facts.

**Final sequence generation**. We draw sequences $\mathbf{x}$ that combine facts with templates. First, we select a template $n$ with (say) position $\mathbf{I} = (i, j)$, and sample $\mathbf{z} \sim \mathcal{D}_n$. Then, we insert a chosen fact $(a, b)$ in positions $(i, j)$ to form the final sequence $\mathbf{x}$, i.e., $x_i \leftarrow a, x_j \leftarrow b$ and $x_t \leftarrow z_t$ for $t \notin \{i, j\}$.

### 2.1 LEVEL OF DIVERSITY

The **diversity-level parameter** `DIV` $\in (0, 1)$ is such that every fact-pair $(a_k, b_k)$, $k \in [K]$ appears embedded in `DIV`$\cdot N$ different templates during training. Larger `DIV` indicates a higher diversity level. To describe template-fact pairings during training, we define binary *in-distribution (ID) exposure mask* $\mathbf{M}_{\text{in}} \in \{0, 1\}^{N \times K}$ with $\mathbf{M}_{\text{in}}[n, k] = \mathbb{1}\big[(a_k, b_k) \text{ occurs in training in a sequence drawn from } \mathcal{D}_n\big]$. Thus, the $k$-th column lists all templates in which the $k$-th fact appears during training. We set `DIV`$\cdot N$ entries of the $k$-th column of $\mathbf{M}_{\text{in}}$ to 1 uniformly at random. The rest $(1-$`DIV`$)N$ templates are unseen at training for the specific $k$-th fact, forming its OOD set. See Fig. 1-B for an example with `DIV` $= 0.2$.

## 2.2 CONTEXTUAL STRUCTURE

We define contextual structure as the specific configuration of templates, particularly how their two key constituents vary: fact placeholder and statistical contexts. We consider the following setups, all using exactly $N$ templates, to allow their direct comparison across identical DIV values:

**1) Position-varied Structure (MC1POSN).** All templates use the same statistical distribution, i.e., $\mathbf{P}_n = \mathbf{P}$ for all $n \in [N]$, while fact placeholders $\mathbf{I}_n$ vary across templates $n$.

**2) Statistical-varied Structure (MCNPOS1).** Each template has its own distinct transition matrix $\mathbf{P}_n$, while using the same fact placeholders ($\mathbf{I}_n = \mathbf{I}$ for all $n \in [N]$).

**3) Mixed-varied Structure (MCNPOSN).** Each template has its own distinct transition matrix $\mathbf{P}_n$ and distinct position placeholder $\mathbf{I}_n$.

## 2.3 EVALUATION METRICS: IN- AND OUT-OF-DISTRIBUTION PERFORMANCE

To systematically evaluate the model, we measure its generalization using controlled probing. For sequence $\mathbf{x}$, drawn from template $n \in [N]$, that contains a given fact pair $(a, b)$, we prompt the model with its first half, $\mathbf{x}_{1:T/2}$, containing the source token $a$, and have it auto-regressively complete the second half, i.e., $\hat{\mathbf{x}}_{T/2+1:T}$. A successful completion requires the model to simultaneously demonstrate: (1) **structural understanding**, by placing a fact token only at the designated position $j_n$ and generic tokens elsewhere; (2) **factual recall**, by generating the correct target token $b$ in the completion; and (3) **statistical consistency**, by ensuring the generated generic tokens follow the template's statistical rules.

We quantify each of these aspects with the following metrics. For positional and factual evaluation, we focus on simple zero-one accuracy in the main text and defer entropy-based counterparts to App. B. For each metric, we distinguish between in-distribution (ID) and out-of-distribution (OOD) performance: We evaluate the model on sequences from two disjoint sets of template-fact pairs $(n, k)$: those seen during training (ID, where $\mathbf{M}_{\text{in}}[n, k] = 1$) and those held out during training (OOD, where $\mathbf{M}_{\text{in}}[n, k] = 0$).

1. **Position accuracy** assesses learning the composition of the two streams: placing a fact token from $\mathcal{V}_\mathcal{K}$ *only* at the designated target position and generic tokens $\mathcal{V}_\mathcal{D}$ everywhere else. Formally:
$$\mathbf{Acc}_{\text{pos}} := \tfrac{1}{2}\big(\mathbb{1}[\hat{x}_{j_n} \in \mathcal{V}_\mathcal{K}] + \tfrac{1}{T/2-1}\sum_{t>T/2, t\neq j_n} \mathbb{1}[\hat{x}_t \in \mathcal{V}_\mathcal{D}]\big). \tag{1}$$

2. **Factual accuracy** measures factual recall, requiring that the only fact token in the completion be the correct target $b$. Concretely,
$$\mathbf{Acc}_{\text{fact}} := \mathbb{1}\big[\{\hat{x}_t\}_{t>T/2} \cap \mathcal{V}_\mathcal{K} = \{b\}\big]. \tag{2}$$

3. **Statistical loss** evaluates statistical generalization. We compute the KL divergence between the model's predictions and the ground-truth, averaged over the set of positions $\mathcal{G}$ where a generic token is generated. For each $t \in \mathcal{G}$, we compare the model's distribution over generic tokens (after masking the fact tokens), $\mathbf{p}_t \in \Delta^{V_\mathcal{D}}$, against the ground-truth $\mathbf{p}_t^*$, the row in the template's transition matrix $\mathbf{P}_n$ specified by the previous generic tokens in the sequence.
$$\mathbf{Loss}_{\text{stat}} := \tfrac{1}{|\mathcal{G}|} \sum_{t\in\mathcal{G}} \text{KL}\big(\mathbf{p}_t \,\|\, \mathbf{p}_t^*\big), \tag{3}$$

## 3 RESULTS

**Experimental setup**. We use a 4-layer decoder-only Transformer, (Radford et al., 2018) trained auto-regressively with the standard next-token prediction loss. Each training sequence has length $T = 50$. We use a template pool of size $N = 10$ and a fact set of $K = 100$ source-target pairs. For the MC, we choose a generic vocabulary set of size $V_\mathcal{D} = 3$. We sweep diversity DIV from $0.1 - 0.9$ and train the model for $30k$ iterations. Unless otherwise noted, metrics are averaged over three random initializations over both model initialization and data generation (including template definition and ID/OOD splits). For each template configuration of Sec.2.2, $\mathbf{Loss}_{\text{stat}}$, $\mathbf{Acc}_{\text{pos}}$ and $\mathbf{Acc}_{\text{fact}}$ are reported in Fig. 2. Each figure shows the heatmap of the metrics across a grid of diversity levels DIV $\in (0, 1)$ and training iterations, for sequences randomly drawn from ID/OOD template-fact distributions. In App.C.2, we discuss the impact of model size by showing similar results on 1- and 10-layer models.

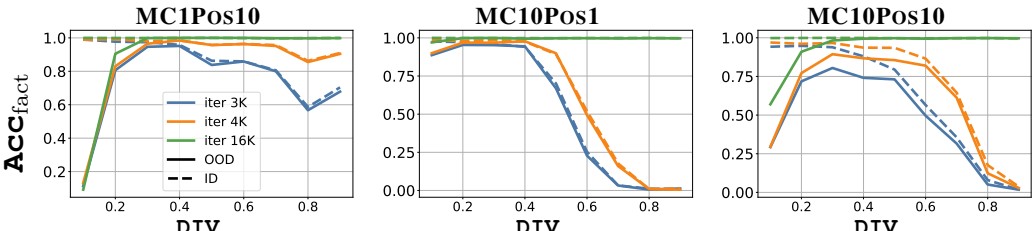

Figure 3: **Diversity pays off with time.** Factual accuracy versus diversity level after 3K, 4K and 16K iterations. Depending on the template structure, very low diversity can remain unrecoverable; no amount of additional training can restore OOD performance. High-diversity runs begin with lower accuracy but continue to improve until they surpass low-diversity models, demonstrating that diversity incurs an initial cost yet yields long-term benefits. Each bold curve is the average over three individual runs for OOD (solid) and ID (dashed) data.

## 3.1 IMPACT OF DIVERSITY ON IN-DISTRIBUTION PERFORMANCE

Across contextual structures and full range of diversity levels, the model ultimately reaches perfect ID performance on all three metrics. Yet, the rate varies. As Figs. 2-(a,b) show, both $\textbf{Loss}_{\text{stat}}$ and $\textbf{Acc}_{\text{pos}}$ converge to 0 and 1, respectively at nearly the same rate, largely unaffected by DIV. In contrast, $\textbf{Acc}_{\text{fact}}$ is sensitive to diversity: the more templates in which a fact appears in training, the longer the model requires to disentangle the correct source–target mapping from contextual patterns. Fig. 2-(c) illustrates this delay clearly: as DIV increases (lower rows in heat map), the yellow band marking perfect recall shifts rightward, signifying that additional training is needed to reach full accuracy in higher-diversity settings. However, the severity of this slowdown depends on the contextual structure, with the MC1POS10 setup showing only a minimal effect.

## 3.2 IMPACT OF DIVERSITY ON OUT-OF-DISTRIBUTION PERFORMANCE

On unseen template-fact distributions, the effect of diversity level depends critically on contextual structure: how fact positions and token statistics vary across templates.

**Position accuracy (Fig. 2-(b)).** In the MC10POS1 setup, where all templates share the same placeholder positions, the model separates generic and fact positions at nearly the same rate for every diversity level DIV. In the MC1POS10 and MC10POS10 setups, where there are more variations across templates on the position assignments, diversity plays a crucial role: In extreme low-diversity settings (i.e., DIV = 0.1 or 0.2), progress in $\textbf{Acc}_{\text{pos}}$ slows down noticeably or fails. However, once diversity is moderate or high, the learning speed differences across DIV become negligible.

**Statistical loss (Fig. 2-(a)).** In the MC1POS10 setup, where all templates share the same token statistics (i.e., identical transition matrices $\mathbf{P}_n$), the statistical loss is largely unaffected by the diversity level and converges at a consistent rate regardless of DIV. In contrast, for the MC10POS1 and MC10POS10 setups, which feature statistical variations across templates, diversity becomes a critical factor. At low diversity levels (DIV < 0.3), prolonged training causes the model's predicted distribution to diverge from the ground truth, as displayed by the light bars in the top-right corners of the OOD heatmaps in Fig. 2-(a)

**Factual accuracy (Fig. 2-(c)).** The effect of diversity on factual recall is more nuanced. At DIV = 0.9, $\textbf{Acc}_{\text{fact}}$ improves more slowly, just as it does on ID data, whereas at DIV $\leq$ 0.2 the model suffers severe factual errors. The severity of this low-diversity failure, however, depends on the contextual structure. In MC1POS10 setup, where all templates share the same transition matrix, the slowdown under high diversity is modest, yet the low-diversity failure is severe. In contrast, in MC10POS1, low diversity slightly delays reaching high $\textbf{Acc}_{\text{fact}}$ rather than causing complete failure, as the fixed position signal across templates acts as a strong positional cue, making the model less sensitive to the lack of contextual variety. Lastly, the behavior for MC10POS10 is a hybrid of MC10POS1 and MC1POS10: It fails at low diversity and shows slower convergence at high diversity.

**Diversity trade-off for factual accuracy (Fig. 3).** To understand how the impact of diversity on factual recall depends on training duration, we plot $\textbf{Acc}_{\text{fact}}$ versus diversity level DIV at three training checkpoints – roughly 3K, 4K, and 16K iterations. With a short budget (3K steps),

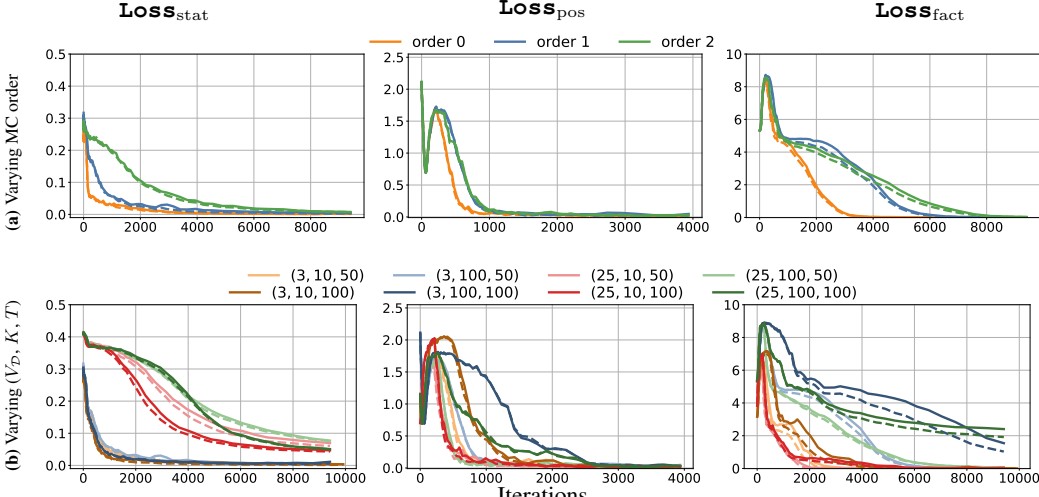

**Figure 4: The complexity of one stream impacts learning of the other.** Speed of learning the statistical, position, and factual metrics as data complexity is varied in a high-diversity setting, by varying **(top)** the MC order, and **(bottom)** the generic vocabulary size $V_{\mathcal{D}}$, number of facts $K$, and sequence length $T$. See Sec.3.3 for discussion and App.B for metrics.

*intermediate* diversity is best: each fact appears in enough contexts to isolate the associations between fact tokens, yet the model is not overloaded with context variations that would stall learning. With more updates (4K steps), factual recall in the high-diversity regimes keeps improving, and in the later checkpoints (16K steps) the performance in high diversity setting recovers and the extra training time compensates for its slower start. Intermediate diversity also benefits from longer training, but extremely low diversity does not recover: except in the MC10Pos1 setup, that has a strong position signal for the facts, low-diversity runs fail to make significant progress with additional training time. Thus, the sweet spot of diversity depends on both training length and how the statistical and factual streams interact in each template. Finally, when low diversity hurts OOD recall (solid lines), the ID recall (dashed lines) remains nearly perfect. However, in high-diversity runs, poor OOD performance is accompanied by similarly poor ID performance, suggesting that training length, not the distribution shift, is the limiting factor.

## 3.3 INTERPLAY BETWEEN STREAMS' COMPLEXITY AND LEARNING DYNAMICS

Here, we study how learning dynamics are affected by systematically varying the complexity of the two streams. We conduct our analysis in a high-diversity regime (`DIV = 0.8`), where all three metrics are learnable with sufficient training. We track $\text{Loss}_{\text{stat}}$ along with $\text{Loss}_{\text{pos}}$ and $\text{Loss}_{\text{fact}}$, the entropy-based analogues of position and factual accuracy (see App. B), while varying four key factors: the generic vocabulary size ($V_{\mathcal{D}} = 3, 25$), the number of facts ($K = 10, 100$), the sequence length ($T = 50, 100$), and the order of the MC (0 to 2). Results are shown in Fig. 4. We observe a strong dependence between how fast each metric can be learned. Specifically:

$\text{Loss}_{\text{stat}}$. As expected, increasing the complexity of the statistical stream slows the decay of $\text{Loss}_{\text{stat}}$. This effect is clear when either increasing the order of the MC (Fig. 4-(a)) or increasing the generic vocabulary size from $V_{\mathcal{D}} = 3$ to $V_{\mathcal{D}} = 25$ for both sequence lengths (Fig. 4-(b)). However, this is not the only factor, as the increased complexity of the factual stream (a larger $K$) also negatively impacts learning, albeit to a lesser degree. Interestingly, we observe a slight acceleration in statistical learning with increased sequence length ($T$). This effect is likely because longer sequences provide more training examples of the short-range dependencies due to the Markovian assumption.

$\text{Loss}_{\text{pos}}$. The speed of positional learning is governed by two factors: sequence length ($T$) and the relative sizes of the vocabulary sets $V_{\mathcal{D}}$ and $V_{\mathcal{K}}$. Longer sequences generally delay the model's ability to identify the fact positions, an effect visible in Fig. 4-(b) by comparing the slower convergence of the darker curves (longer sequences) to the lighter ones. However, when the factual vocabulary is small and comparable to that of the statistical stream (e.g., $K = 10$ vs. $V_{\mathcal{D}} = 25$), the model learns the fact positions rapidly even with longer sequences, as demonstrated by the red curve in Fig. 4-(b).

**Loss**$_{\text{fact}}$. The dynamics of factual learning are more nuanced, showing stronger dependencies on the other two learning aspects. First, factual learning is contingent on positional learning; the model learns *where* to place facts before it learns *what* facts to generate, consistent with observations in Zucchet et al. (2025). Consequently, any factor that slows positional learning, such as increasing the sequence length, slows down factual learning and often has an amplified negative impact. Second, factual learning is not independent of the statistical stream. Even when other factors are held constant, increasing only the statistical stream's complexity (larger $V_{\mathcal{D}}$ or higher order) can also slow down the model's ability to learn facts.

## 4 LOCALIZING THE OPTIMIZATION BOTTLENECK

We have seen that generalization to sequences with unseen template-facts is highly dependent on pretraining diversity level: While models trained with sufficient diversity generalize well across all three metrics of Sec. 2.3, those trained on low-diversity data consistently fail to find a generalizing solution. Note that this failure is not due to a lack of model expressivity since generalization of the model trained on diverse data proves that the generalizing solution is well within the model's capacity. This observation rather points to an optimization bottleneck. More concretely, consider the generalizing parameters $\theta_{\text{hi}}$ learned on a high-diversity dataset. This solution also achieves minimal loss on any low-diversity training set, making it a global minimizer of the low-diversity training objective. However, training directly on low-diversity data, the optimization process converges to a different minimizer, $\theta_{\text{low}}$, which merely performs well on the seen template-facts. This indicates that the low-diversity loss landscape has multiple global minima, and the optimization process is biased towards finding non-generalizing ones (see also App. D for a more formal discussion).

To probe this optimization bottleneck, we re-train the models from scratch across various data diversity levels while applying controlled interventions on a selected module of the model, providing it with respective well-trained weights from $\theta_{\text{hi}}$ which we keep frozen. We freeze the following four modules: (**1**) token embeddings (**E**), (**2**) attention (**Attn**), (**3**) MLP (**MLP**), and (**4**) unembedding layer (**U**). For attention, for each sequence and at each layer, we substitute the pre-computed attention patterns from $\theta_{\text{hi}}$ during each forward pass. We freeze the transferred parameters since we find that a model initialized at $\theta_{\text{hi}}$ and trained on low-diversity data causes the parameters to move away from the set generalizing solutions. (See Fig. 6-(a)). We focus on the MC10POS10 setup which exhibits failures in both factual and statistical modes. While we present results across all diversity levels in Fig. 9, here, we only discuss the most interesting low-diversity setting $\mathtt{DIV} = 0.1$ and copy parameters from $\mathtt{DIV} = 0.9$.

Fig. 5-(a) summarizes the model's performance on statistical, positional, and factual metrics after $\sim 10$K training iterations while applying different types of intervention. Starting with the MLP, we note that freezing its weights to those from $\theta_{\text{hi}}$ shows no improvement on any aspect of generalization. However, interventions on the other modules reveal significant effects:

1. **Unembedding helps with factual recall and restores training-speed consistency.** Providing the model with the unembedding layer from $\theta_{\text{hi}}$ exclusively improves factual recall, resulting in a 20% accuracy increase in this low-diversity setting. However, its most significant impact is on the speed of convergence at higher diversity levels. As shown in Fig. 5-(b), and consistent with the trade-offs discussed in Sec. 3, training the full model end-to-end leads to a progressive slowdown in fact learning at higher diversity levels $\mathtt{DIV} \in [0.5, 0.9]$. Remarkably, with the unembedding frozen, this slowdown vanishes: the model learns facts at a consistent pace across diversity levels.

2. **Attention and token embeddings boost position and statistical accuracy.** Conversely, intervening on the attention mechanism or token embeddings exclusively benefits statistical and structural learning. Providing the attention patterns from $\theta_{\text{hi}}$, effectively telling the model where to focus at each position, allows the model to achieve perfect ($100\%$) **Acc**$_{\text{pos}}$ and a significantly lower **Loss**$_{\text{stat}}$. Similarly, freezing the token embeddings reduces the statistical loss to nearly zero while also yielding near-perfect **Acc**$_{\text{pos}}$.

3. **Combining interventions yields perfect generalization.** With the combination of both types of interventions in (1) and (2), the model leverages both benefits simultaneously. Freezing both the token embeddings and the unembedding layer together results in perfect statistical learning, near-perfect positional accuracy, and a substantial improvement in factual recall of over 50%. The effect is even more pronounced when combining the attention and unembedding interventions; this

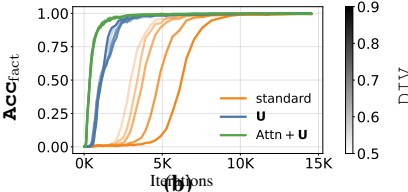

**(a)**

**Figure 5: Tracing the optimization failure to individual model components.** Models trained with a selected module (MLP, unembedding (**U**), token embedding (**E**), or attention (**Attn**)), replaced by its frozen counterpart from a well-trained model on high-diversity data. *Standard* refers to training the model, end-to-end with no interventions. All reported values are averaged over three independent runs. **(a)** Performance on all metrics after 10K iterations on low-diversity (DIV = 0.1) data. Note that for **Loss**$_{stat}$ darker and for **Acc**$_{pos}$ and **Acc**$_{fact}$ brighter colors are optimal. **(b)** Factual accuracy over time, for different diversity values (wiht darker curves corresponding to higher diversity), showing the convergence speed to perfect recall for various interventions. See Sec. 4 for details and App. C.4 for more results.

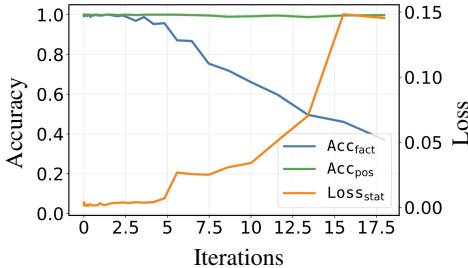 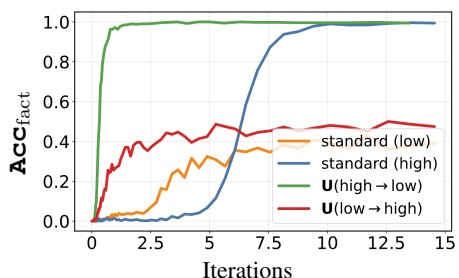

**Figure 6: (left)** Training a model on low-diversity data, even when initialized to high-diversity parameters $\theta_{hi}$, yields poor performance. **(right)** Feature probing experiments in Sec. 4.1. Reference end-to-end training dynamics on high and low-diversity data (blue and orange) vs retraining the unembedding layer **U** with high/low diversity data on features learned from the low/high diversity data (red/green).

setup, along with freezing all three components, leads to near-perfect performance across all metrics. It is important to note that even in this last case, the learning task is not trivial, as the model must still learn all the intermediate computations within the MLP layers, which remain fully trainable.

## 4.1 BEYOND THE UNEMBEDDING LAYER: FEATURES' QUALITY FOR FACTUAL RECALL

We have seen that joint optimization of unembeddings and internal features creates a challenging optimization landscape under low-diversity, impeding finding a generalizing minimizer. This, however, raises a deeper question about the nature of this bottleneck: Is the unembedding layer itself difficult to optimize, or is its challenge a symptom of receiving poor-quality hidden representations from the rest of the model? To isolate the impact of feature quality, we now conduct a series of feature probing experiments where we freeze the main model and exclusively retrain the final unembedding layer.

**Good unembedding cannot fix poor representations**. We freeze all parameters of a model ($\theta_{low}$) trained on data with DIV = 0.1 except for the unembedding, which we retrain under on sequences with high-diversity (DIV = 0.9). Fig. 6-b (red curve) shows this model fails to achieve high factual accuracy, performing only marginally better than the original $\theta_{low}$. Thus, last-layer representations of the low-diversity model are themselves insufficient for generalization, even when unembedding is trained on high-diversity.

**High-quality features enable learning from low-diversity data**. In the reverse experiment, we freeze the body of a high-diversity model $\theta_{hi}$ and retrain its unembedding layer on low-diversity data. This configuration generalizes successfully (Fig. 6-b, green). This finding is twofold. First, it demonstrates that the representations from $\theta_{hi}$ effectively differentiate between fact and template information, allowing a linear classifier to generalize even from low-quality training signals. Second, this outcome contrasts with the degradation seen when training the entire model initialized at $\theta_{hi}$ on low-diversity data (Fig. 6-a). The comparison makes it clear that performance degradation does not

only stem from learning a poor unembedding layer, but also from the internal features themselves degrading when they are not frozen.

Additional investigations on the inner representations and their clustering properties are in App E. Our mechanistic analysis reveals that higher diversity encourages the model to learn internal representations that are less dependent on the prompt's template structure, thereby improving generalization. Moreover, to analyze the impact of data diversity on OOD generalization for factual recall, we construct a minimal experimental setting (see Appendix D for details).

## 5 CONCLUSION AND LIMITATIONS

We have introduced a synthetic framework to systematically investigate the generalization abilities of language models pre-trained on sequences that combine statistical patterns and factual information. The simplicity of our testbed is intentional, offering several advantages: **(1)** it enables fine-grained control over diversity level, while independently varying contextual structure; **(2)** it requires minimal computational overhead; and **(3)** it captures different aspects of language model generalization in both ID and OOD settings. Using this, we have uncovered several illuminating behaviors regarding the interplay between statistical and factual generalization as functions of diversity and structure.

Our findings, derived from this controllable and inexpensive testbed, open several avenues for deeper investigation into the learning dynamics of language models often obscured in large-scale regimes. Building upon our results in Sec. 4, future works can formalize our observations, such as how data diversity shapes the optimization landscape and how different modules come to serve specialized roles. Second, the framework's flexibility welcomes various extensions to study specific phenomena, such as the impact of imbalanced fact exposure and the potential of fine-tuning with specific template formats (mirroring (Allen-Zhu and Li, 2023)). Moreover, this testbed could prove valuable for investigating the limitations and potential advantages of architectures beyond the Transformers considered in this work Finally, the minimalist nature of our testbed was a deliberate choice to enable controlled and low-cost analysis. A key challenge for future work is navigating the trade-off between this analyzability and linguistic realism. While bottom-up refinements are necessary to better capture the complex structures of natural language, it is crucial to maintain a balance that allows for the clear, mechanistic insights that simple testbeds provide.

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

**Table 1:** Table of notations.

| Symbol | Meaning |
| --- | --- |
| $T$ | Sequence length. |
| $\mathbf{x} = (x_1, \ldots, x_T)$ | Token sequence produced by mixing two streams. |
| $\mathcal{V} = [V]$ | Global vocabulary. |
| **Factual stream** | |
| $K$ | Number of atomic facts. |
| $\mathcal{K} = \{(a_k, b_k)\}_{k=1}^K$ | Set of $(\mathrm{source}, \mathrm{target})$ pairs. |
| $\mathcal{A} = \{a_k\}, \ \mathcal{B} = \{b_k\}$ | Source and target vocabularies. |
| $\mathcal{V}_K = \mathcal{A} \cup \mathcal{B}$ | Fact vocabulary. |
| **Statistical stream (templates)** | |
| $N$ | Number of templates. |
| $\mathcal{D}_n$ | Distribution associated with template $n$. |
| $\mathcal{V}_\mathcal{D} = \mathcal{V} \setminus \mathcal{V}_K$ | Set of generic vocabularies (the MC vocabulary). |
| $\mathbf{I}_n = (i_n, j_n)$ | Placeholder positions for fact pairs in template $n$ ($i_n \leq T/2 < j_n$). |
| $\mathbf{P}_n$ | The MC transition matrix associated with template $n$. |
| **Exposure and diversity** | |
| $\mathbf{M}_{\mathrm{in}} \in \{0,1\}^{N \times K}$ | ID exposure mask: $(n, k)$ entry is 1 if fact $k$ occurs in template $n$ during training. |
| $\mathtt{DIV} \in (0, 1]$ | Diversity level: fraction of templates in which each fact appears. |
| **Metrics** | |
| $\mathbf{Acc}_{\mathrm{pos}}$ | Structural (position) accuracy (Eq. (1)). |
| $\mathbf{Acc}_{\mathrm{fact}}$ | Factual accuracy (Eq. (2)). |
| $\mathbf{Loss}_{\mathrm{stat}}$ | Statistical loss (Eq. (3)). |
| $\mathbf{Loss}_{\mathrm{pos}}$ | Entropy analogue of $\mathbf{Acc}_{\mathrm{pos}}$ (Eq. (5)). |
| $\mathbf{Loss}_{\mathrm{fact}}$ | Entropy analogue of $\mathbf{Acc}_{\mathrm{fact}}$ (Eq. (7)). |

**Other notations.** We denote matrices/vectors/scalars as $\mathbf{A}/\mathbf{a}/a$ respectively. We view token-sequences as vectors and denote $\mathbf{a}_{t_1:t_2}$ as the subsequence from index $t_1$ to $t_2$. We denote $\mathbf{A}[i, j]$ the $(i, j)$-th entry of matrix $\mathbf{A}$, and respectively for vectors. We let $\mathbb{S}(\cdot)$ denote the softmax map, $\Delta$ the simplex, and $\mathrm{KL}(\mathbf{p}_1 \,\|\, \mathbf{p}_2)$ the Kullback–Leibler (KL) divergence between probability vectors $\mathbf{p}_1, \mathbf{p}_2$. We denote $[N] := \{1, \ldots, N\}$ and use $\mathbb{1}[\mathcal{C}]$ for the indicator function (1 if condition $\mathcal{C}$ is satisfied, 0 otherwise).

**Overview of supplementary material**. Section A expands on the literature discussion in Section 1. Section B introduces additional metrics used in Section 3.3. Section C provides additional experimental details and results to complement Sections 3 and 4. Section D introduces a minimal toy setting – as a starting point for theoretical analysis. Finally, Section E examines the model's internal embeddings and how training diversity affects them. Code available at `https://anonymous.4open.science/r/MCPos-BCB1/README.md`

# A ADDITIONAL DETAILS ON RELATED WORKS

LLMs have been observed to pack a substantial amount of knowledge in their parameters during pretraining, allowing them to answer real-world questions without consulting external resources (Petroni et al., 2019; Roberts et al., 2020), raising the question of whether they can replace conventional knowledge bases (Omar et al., 2023; Sun et al., 2023). By inspecting the internal activations of different modules and layers, a growing body of mechanistic interpretability work seeks to understand where LMs store knowledge (Meng et al., 2023; Dai et al., 2021; Geva et al., 2020) and how they recall it (Geva et al., 2023; Lv et al., 2024; Choe et al., 2025). Other studies trace each learned fact back to the pre-training data to investigate which corpus patterns enable its acquisition (Elazar et al., 2022; Akyürek et al., 2022; Li et al., 2022) and demonstrate that recall accuracy depends on the number of exposures to that fact in the pretraining corpus (Kandpal et al., 2023; Allen-Zhu and Li, 2023). Studies on learning dynamics similarly find that different knowledge

types are learned at different rates (Liu et al., 2021). Chang et al. (2024) probe the factual recall dynamics by injecting fictional facts during pre-training and tracking their probabilities over time, observing that knowledge accumulates through many small "micro-updates" that gradually decay unless the fact is periodically reinforced to avoid forgetting.

Recent works have initiated systematic exploration of factual recall in language models using *controlled synthetic setups*. Typically, these works model each fact as a triplet (source, *relation*, target) embedded in a context; the model is then probed with a context containing a (source, relation) and must produce the corresponding target (Allen-Zhu and Li, 2023; Nichani et al., 2024; Zucchet et al., 2025). We adopt a similar framework but omit the *relation* token. The simpler (source, target) setup suffices for our purpose of studying how models acquire and recall factual associations along with other aspects of generalization from first principles.

Allen-Zhu and Li (2023) use a controlled synthetic biography dataset: each biography entry is a multi-sentence paragraph about an individual, the *source*, where each sentence represents a (relation, target) chosen from a set of fixed-sentence templates, e.g., "*<Name> was born in <City>*". They specifically consider question-answer (QA) formats for probing knowledge, e.g., "*What is <Name>'s city of birth? <City>*", which are shown at training time for only a subset of individuals. We treat these QA forms as just another template family and evaluate by probing the model with any template that was *unseen* for a given fact during training. Focusing on factual recall performance, they show that factual recall after fine-tuning on question templates is only possible if facts are observed in several templates during pre-training and more paraphrase diversity markedly boosts recall. In other words, successful recall requires *varied* exposure to each fact, not mere repetition in a single template.

Our data-generation scheme extends this core insight by further abstracting the template–fact structure. This provides fine-grained control over the statistical and structural composition of the data, aspects that were fixed in the original work. We focus exclusively on pre-training experiments here, although the same abstract framework could also be used to study fine-tuning. Yet, we show that even without finetuning and even in minimal settings like those described in Sec. D low-diversity can impair generalization. Our analysis of the impacts of diversity on the statistical aspects of generalization as well as the systematic categorization of different contextual structures are also unique to our study compared to this prior work.

Recent work by Zucchet et al. (2025) analyzes the dynamics of factual recall in a synthetic biography setup, similar to that of Allen-Zhu and Li (2023), and reports a stage-wise dynamic. By inspecting the model's predictions at the target position across training checkpoints, the model first restricts its choice to the fact vocabulary and only later learns the correct mapping from the the specific (source, relation) present in the context to the correct fact token. We make similar observation between factual and position accuracy Sec. 3.3. Our evaluation, however, is broader: we probe the model with an incomplete prompt and grade the *entire* completion from different aspects – whether the correct fact token appears in the correct position, whether the remaining positions are filled with generic tokens, and whether those tokens follow the template's statistical pattern. As Zucchet et al. (2025) follow similar setup as Allen-Zhu and Li (2023), the unique and distinctive characteristics of our setup mentioned above—particularly the explicit statistical stream enabling a joint study of statistical and factual generalization aspects, alongside systematic control over contextual structure and diversity—apply equally as differentiators here.

Both of the above referenced studies also vary the data distribution, exploring how "celebrity" entries – individuals whose biographies appear in many templates during pre-training – affect factual recall for less-frequent entries and alter learning dynamics. While our current work focuses only on overall context diversity, our flexible framework can readily accommodate such experiments on the impact of data distribution by appropriately designing the template-fact exposure matrix $M_{in}$ to vary fact frequencies across templates. We leave this as interesting future work.

Perhaps the most closely-related work in terms of model abstraction, although coming from differing motivations, appears in Nichani et al. (2024). While they focus on capacity and storage tradeoffs for factual recall, we investigate the impacts of diversity and tradeoffs between statistical and factual accuracy. In their synthetic data setup, they sample the (source, relation, target) mappings randomly by choosing them from a fact set. Each sequence is generated by placing the (source, relation) at two random position, appending the target at the end and filling the remaining positions with tokens uniformly drawn from a disjoint *noise* vocabulary (functionally identical to our generic tokens).

Training then minimizes the loss at the target position, focusing on the fact storage capacity of the model. Similarly in our setup, we preserve the separation of facts from background tokens but introduce a *structured* statistical stream: generic tokens are generated by a Markov process and fact placeholders occupy slots that can vary across different templates. This richer design allows us to introduce and systematically investigate how contextual structure and diversity affect performance, while extending the analysis beyond factual recall to include statistical and structural generalization, aspects not explored by Nichani et al. (2024). Through our effort to identify minimal toy settings where key tradeoffs, such as the impact of diversity on factual recall, are maintained, it might be possible to leverage some of the theoretical ideas from Nichani et al. (2024) to analyze our findings. However, this would require various non-trivial extensions, particularly incorporating the impact of diversity and adapting for an autoregressive generation setting rather than their last-token prediction.

We also review a growing body of recent works that have used Markov chains, as we do here to model the statistical stream, to study various behavioral aspects of transformers in next-token prediction tasks. Makkuva et al. (2024a) study the loss landscape properties of a single-layer transformer trained on sequences drawn from a fixed order-1 Markov chain, characterizing the influence of transition probabilities and architectural choices on the loss landscape. Edelman et al. (2024b) demonstrates that transformers trained on sequences generated from random order-1 Markov chains develop the ability to perform in-context inference on unseen Markov chains by outputting bigram statistics inferred from the context. Park et al. (2025) extend this framework by examining the regime where training sequences are drawn from a fixed, finite set of Markov chains. Rajaraman et al. (2024) has analyzed the representational capacity of transformers for in-context learning of order-$k$ Markov chains. None of these works combines MCs with factual information, as we do here. More broadly, synthetic tasks have proven valuable for dissecting LM behavior beyond our focus, including self-attention mechanisms Tian et al. (2023); Li et al. (2024); Ren et al. (2024); Makkuva et al. (2024b), in-context learning (Garg et al., 2022; Bietti et al., 2023; Edelman et al., 2024a; Park et al., 2025), and distributional associations Chen et al. (2025).

Additionally, our work is broadly related to prominent neuroscientific theories of language processing. For instance, some theories propose separate cortical circuits dedicated to syntactic and semantic processing, suggesting a functional segregation in the brain for handling language structure versus meaning (Matchin and Hickok, 2020). Other research points to a stage-wise progression in human language comprehension, where an initial, rapid phase of syntactic analysis is thought to precede and scaffold deeper semantic integration (Friederici, 2017).

## B METRIC OVERVIEW

In this section, we review the metrics in Sec. 2.3 along with introducing the entropy-based metrics $\texttt{Loss}_{\text{fact}}$ and $\texttt{Loss}_{\text{pos}}$ used in Sec. 3.3 to study the learning dynamics. Recall that, whether ID or OOD, we measure the model's: (i) adherence to the composition rule between the two streams, (ii) accuracy of factual recall, and (iii) ability to follow the statistical patterns of the background template.

For each ID/OOD template-fact combination $(n, k)$, we draw a sequence $\mathbf{x}_{1:T}$, use its first half $(\mathbf{x}_{1:T/2})$ as a prompt, and have the model generate the second half. We refer to the generated tokens at positions $t > T/2$ by $(\hat{x}_{T/2+1}, \ldots, \hat{x}_T)$. Let $\hat{\boldsymbol{\ell}}_t(\cdot) \in \mathbb{R}^V$ be the model's predicted *logits* at position $t \leq T$ conditioned on input $(x_1, \cdots, x_{T/2}, \hat{x}_{T/2+1}, \cdots, \hat{x}_{t-1})$. Also, let $\hat{\mathbf{p}}_t(\cdot) \in \Delta^{|\mathcal{V}|}$ be the softmax *probability* at this position. Without loss of generality, assume the index of the generic tokens is $[|\mathcal{V}_{\mathcal{D}}|]$.

1. **Position accuracy/loss:** To obey the composition rule between the factual and statistical streams, the generated sequence should contain a fact tokens from $\mathcal{V}_{\mathcal{K}}$ *only* at the target position $j_n$, and all other positions should contain tokens from the generic vocabulary $\mathcal{V}_{\mathcal{D}}$. Formally, we define *position accuracy* as:

$$\texttt{Acc}_{\text{pos}} = \tfrac{1}{2}\big(\mathbb{1}[\hat{x}_{j_n} \in \mathcal{V}_{\mathcal{K}}] + \tfrac{1}{T/2-1}\sum_{t>T/2, t\neq j_n} \mathbb{1}[\hat{x}_t \in \mathcal{V}_{\mathcal{D}}]\big). \tag{4}$$

We also define the *position loss* by measuring how much of the probability distribution at each position is accumulated on the correct subset of the vocabulary. Specifically,

$$\texttt{Loss}_{\text{pos}} = -\tfrac{1}{2}\big(\log(\sum_{v\in\mathcal{V}_{\mathcal{K}}} \hat{\mathbf{p}}_{j_n}(v)) + \tfrac{1}{T/2-1}\sum_{t=T/2+1, t\neq j_n}^{T} \log(\sum_{v\in\mathcal{V}_{\mathcal{D}}} \hat{\mathbf{p}}_t(v)))\big). \tag{5}$$

2. **Factual accuracy/loss:** We define *factual accuracy* as the existence of the correct prediction of the target in the prompt completion generated by the model:

$$\texttt{Acc}_{\text{fact}} := \mathbb{1}\big[\{\hat{x}_t\}_{t>T/2} \cap \mathcal{V}_{\mathcal{K}} = \{b_k\}\big]. \tag{6}$$

We accordingly define the *factual loss* as

$$\texttt{Loss}_{\text{fact}} := -\log\left(\hat{\mathbf{p}}_{j_n}(b_k)\right). \tag{7}$$

3. **Statistical loss:** Denote $\mathcal{G} \subseteq [T/2+1, T]$ the set of positions in the generated completion that are filled with generic tokens from $\mathcal{V}_{\mathcal{D}}$. For each such position $t$, we compare the model's distribution over generic tokens with the ground-truth MC distribution ($\mathbf{P}_n$) of the template. Concretely, keep the first $|\mathcal{V}_{\mathcal{D}}|$ coordinates of the logit $\tilde{\boldsymbol{\ell}}_t = \boldsymbol{\ell}_t[\mathcal{V}_{\mathcal{D}}]$ corresponding to the generic tokens and compute the model's distribution over $\mathcal{V}_{\mathcal{D}}$ as $\tilde{\mathbf{p}}_t = \mathbb{S}(\tilde{\boldsymbol{\ell}}_t)$. Let $\mathbf{p}_t^*$ be the row of $\mathbf{P}_n$ that corresponds to the preceding generic token of $\hat{x}_t$. We measure *statistical loss* as:

$$\texttt{Loss}_{\text{stat}} := \frac{1}{|\mathcal{G}|} \sum_{t \in \mathcal{G}} \text{KL}\left(\tilde{\mathbf{p}}_t \,\|\, \mathbf{p}_t^*\right). \tag{8}$$

## C  ADDITIONAL EXPERIMENT RESULTS

### C.1  EXPERIMENT DETAILS OF SEC. 3

In the experiments of Secs. 3, we use a decoder-only Transformer (Radford et al., 2018) with 4 layers and 4 heads trained auto-regressively with the standard next-token prediction loss. Unless stated otherwise, each training sequence has length $T = 50$. We use a template pool of size $N = 10$. We choose a generic vocabulary set of size $V_{\mathcal{D}} = 3$ for the MC and a fact set of $K = 100$ source-target pairs, which gives a total of $V = 203$ vocabularies. We allocate the first $2K$ tokens in the vocabulary to the fact stream and build up the fact set $\mathcal{K}$ by randomly pairing these tokens together. We build the statistical stream by generating $N$ random transition matrices (sampled from the Dirichlet distribution with parameter $\mathbf{1}$) for each template, and $N$ distinct position placeholders $\mathbf{I}_n = (i_n, j_n)$ along the sequence. We train the models for $30k$ iterations and a batch size of 64, with AdamW (Loshchilov and Hutter, 2017) and a fixed learning rate of $10^{-3}$.

Unless otherwise specified, all reported metrics are averaged over three independent runs, each with different random seeds for both data generation (which controls the random generation of $\mathcal{K}$, $\{\mathbf{P}_n\}_{n\in[N]}$ and $\{\mathbf{I}_n\}_{n\in[N]}$) and model initialization. For experiments that sweep across the diversity level `DIV` (e.g., Fig. 2), we fix these seeds for a given run and vary only `DIV` from 0.1 to 0.9.

### C.2  IMPACT OF MODEL SIZE

Here, we repeat our main experiments – originally with 4-layer transformer – this time with 1-layer and 10-layer models. In this section, all experiments are run with a learning rate $10^{-4}$. In Fig. 7, we gather heatmaps of (a) $\texttt{Loss}_{\text{stat}}$, (b) $\texttt{Acc}_{\text{pos}}$, and (c) $\texttt{Acc}_{\text{fact}}$ on OOD sequences for each model size.

Across all model sizes, the impact of diversity level and training duration is identical. For example, in terms of factual recall, low diversity causes a failure, moderate diversity is best for intermediate training length, and high diversity achieves optimal performance only after long training. The primary impact of depth is on the convergence speed: at any fixed iteration count, the 10-layer model achieves higher accuracy than the 4-layer, which in turn outperforms the 1-layer. Notably, larger models help slowly improve *position accuracy* even under low diversity (seen as brighter colors in the upper-right portions of panel b). However, the factual recall failure at extreme low diversity persists across depths, highlighting that model capacity alone cannot substitute for contextual variety in achieving robust OOD generalization.

### C.3  MC10POS10: *Structural*-OOD DATA

The MC10POS10 setup enables us to evaluate a particularly challenging form of generalization beyond standard ID/OOD splits: *structural*-OOD templates that test pure compositional reasoning.

Recall that in MC10POS10, each of the $N = 10$ templates is uniquely specified by a transition matrix and position pair: $(\mathbf{P}_n, \mathbf{I}_n)$ for $n \in [N]$. As in our other contextual structure settings, we

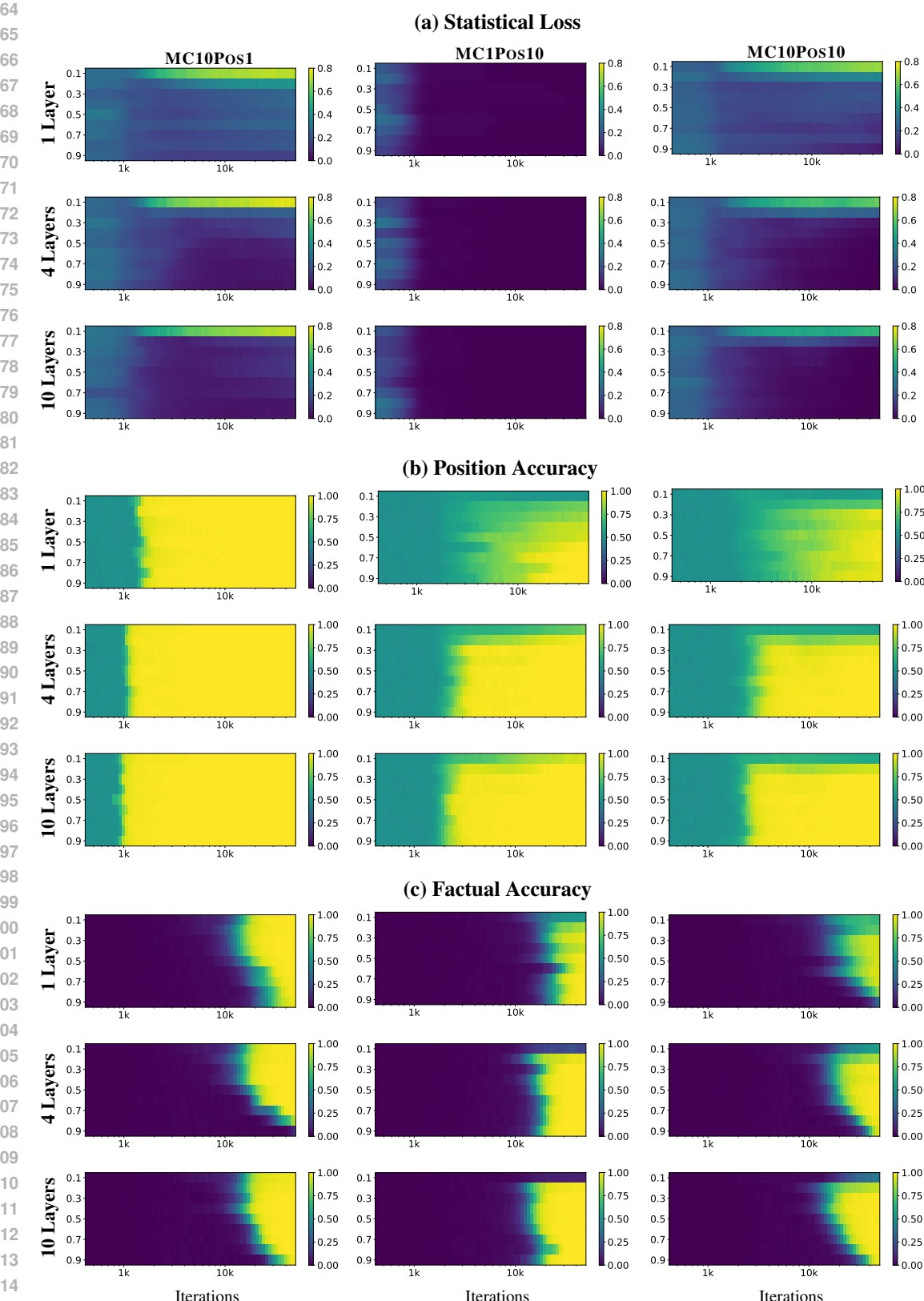

**Figure 7: Impact of model size.** Replication of the experiments in Sec. 3 (4-layer) with smaller (1-layer) and larger (10-layer) models on OOD sequences. With increased model capacity, we need fewer iterations to achieve the same level of performance. However, model capacity alone cannot alleviate the failure of factual recall at low diversity levels.

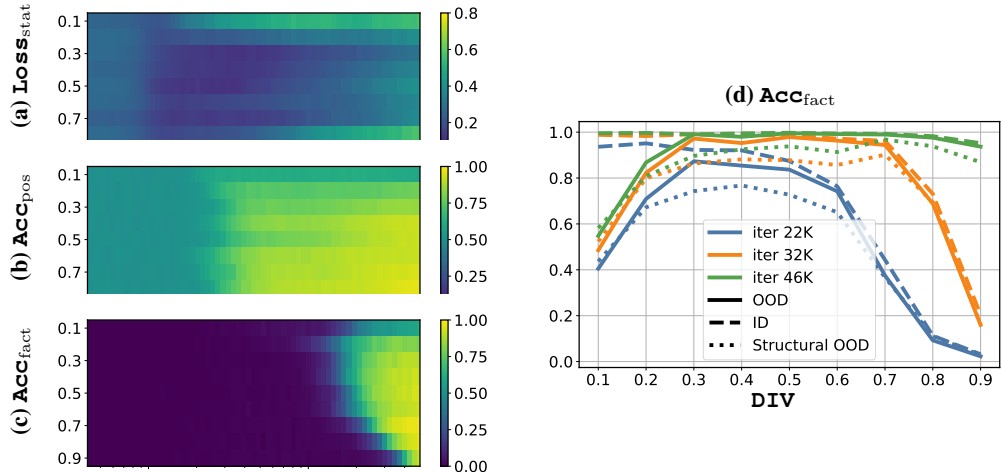

**Figure 8: Structural-OOD performance in the MC10POS10 setup. (a)** $\mathbf{Loss}_{\text{stat}}$, **(b)** $\mathbf{Acc}_{\text{pos}}$ and **(c)** $\mathbf{Acc}_{\text{fact}}$ for the experimental setup of Fig. 2 on sequences drawn from structural-OOD templates, defined in Sec. C.3. Panel **(d)** replots $\mathbf{Acc}_{\text{fact}}$ versus DIV, at three training checkpoints as in Fig. 3.

define ID and OOD templates for a given fact $(a, b)$ based on whether they appeared with this fact pair during training.

The distinctive feature of MC10POS10 is that it allows us to test pure compositional generalization by forming new templates through pairing each $\mathbf{P}_n$ with a position pair $\mathbf{I}_{n'}$, for $n \neq n'$. Specifically, in this case, each mixed template combines a Markov chain and position both familiar in isolation but never jointly encountered, and the model must combine these familiar subcomponents to generalize to the structural-OOD sequences. We call sequences generated from such templates *structural*-OOD sequences.

Fig. 8 plots $\mathbf{Loss}_{\text{stat}}$, $\mathbf{Acc}_{\text{pos}}$ and $\mathbf{Acc}_{\text{fact}}$ on *structural*-OOD sequences in the 4-layer experimental setup of Fig. 7. The qualitative trends mirror our standard OOD findings: Under low diversity levels, factual recall fails catastrophically; mid-training, optimal performance is achieved with intermediate diversity levels; and high diversity helps recovering the performance in long training regimes. However, the absolute value of the metrics remains lower than on the original OOD split, hinting the model needs even longer training to master completely novel template combinations. Position accuracy shows a similar pattern: Sharp failure at low diversity, but little sensitivity to diversity elsewhere. Interestingly, $\mathbf{Loss}_{\text{stat}}$ is hit hardest by these novel template compositions at test time. We leave a deeper investigation for future work, In panel (d), we further visualize, as in Fig. 3, that extended training uncovers the long-term benefits of high diversity, even though it may slow progress during the intermediate phase.

## C.4 EXPERIMENT DETAILS OF SEC. 4

In the controlled experiments of Section 4, we use the same data and model initialization seeds for each experimental run as those used to train the corresponding well-trained model $\boldsymbol{\theta}_{\text{hi}}$. This ensures that the underlying fact and template definitions are identical and that training begins from the exact same initialization point. After initializing the model, we transfer the desired parameter subset from $\boldsymbol{\theta}_{\text{hi}}$, freeze it, and then retrain the model on a new dataset that differs only in its diversity level.

While the main text (Sec. 4 and Fig. 5) focused on the impact of interventions at the lowest diversity level (DIV = 0.1), Fig. 9 provides the complete results across all diversity levels and training iterations in a format similar to Fig. 2. Here, we only focus on the OOD performance. For completeness, the figure also includes results for interventions on other modules that had a limited impact and were therefore not discussed in the main text. Table 12 reports metrics at iteration 10k (mid-training), analogous to the final results shown in Fig. 5-(a). This mid-training snapshot more clearly demonstrates

that intervening on $\mathbf{U}$ accelerates learning, even in low-diversity settings. We replicate the trends in Sec. 4, for a one-layer, one-head Transformer, as detailed in Figs. 10–11.

## D  MINIMAL SETTING TO UNDERSTAND THE IMPACT OF LOW DIVERSITY

Focusing on factual recall, we consider a minimal toy setting with $N$ templates and $K = N$ fact pairs with sequences of length $T = 2 \times N$. For further simplicity compared to our other settings, we let the generic tokens $\mathcal{V}_\mathcal{D}$ to be drawn from uniform distribution over the $V_\mathcal{D} = 3$ tokens. For the source-target pair, we define each template $n \in [N]$ by a position pair $\mathbf{I}_n = (n, n + N)$. We compare the performance on two diversity levels: low diversity $\texttt{DIV} = 1/N$ and high diversity $\texttt{DIV} = (N - 1)/N$.

Fig. 13 reports OOD factual recall in this setup for two diversity levels high (blue) and low (red) in three minimal settings: (a) $N = 3$ with a 1-layer model, (b) $N = 3$ with a 4-layer model, (c) $N = 5$ with a 4-layer model. In all cases ID performance reaches $100\%$ by the end of training, so we only show the OOD results. As seen in panel (a), a 1-layer model is expressive enough to achieve perfect factual recall performance on the task, as it achieves perfect OOD (and ID) factual recall when trained with high diversity. However, with low diversity, the training algorithm fails to find this generalizing solution. Instead it converges to a solution that generalizes for the ID templates, but does not necessarily perform well on OOD templates. Increasing the model capacity roughly helps with the performance in the low-diversity case as shown in panel (b). However, increasing the task complexity by simply increasing $N$, the same large model of panel (b), fails again at finding the generalizing solution.

We can formally think of this failure under low-diversity as follows. Following the notation in Sec.2, the ultimate learning goal is to find model parameters $\boldsymbol{\theta}^*$ that minimize the next-token prediction (NTP) loss over the complete distribution over the choice of the templates and facts, i.e.,

$$\boldsymbol{\theta}^* \in \arg\min_{\boldsymbol{\theta}} \left\{ \mathcal{L}_{\text{tot}}(\boldsymbol{\theta}) := \sum_{k \in [K]} \sum_{n \in [N]} \mathbb{E}_{\mathbf{x} \sim \{\mathcal{D}_n^k\}} \ell_{\text{NTP}}(\mathbf{x}; \boldsymbol{\theta}) \right\},$$

where $\mathcal{D}_n^k$ is the distribution over sequences drawn from the $n$-th template with the fact placeholders filled with the $k$-th fact $(a_k, b_k)$, and $\ell_{\text{NTP}}$ is the NTP loss on sequence $\mathbf{x}$ parameterized by model parameters $\boldsymbol{\theta}$. Note here that the total loss averages over *all* $N$ templates. We assume henceforth that the model is sufficiently expressive such that $\mathcal{L}_{\text{tot}}(\boldsymbol{\theta}^*)$ attains the loss lower bound (over all possible parameterization). This is the case in all our settings.

We can now decompose this loss into two components as $\mathcal{L}_{\text{tot}}(\boldsymbol{\theta}) = \mathcal{L}_{\text{ID}}(\boldsymbol{\theta}) + \mathcal{L}_{\text{OOD}}(\boldsymbol{\theta})$, where $\mathcal{L}_{\text{ID}}(\cdot)$ aggregates the ID templates and the complement $\mathcal{L}_{\text{OOD}}(\cdot)$ term contains the OOD templates for each fact. Concretely, let

$$\mathcal{L}_{\text{ID}}(\boldsymbol{\theta}) := \sum_{k \in [K]} \sum_{n \,:\, \mathbf{M}_{\text{in}}[n,k]=1} \mathbb{E}_{\mathbf{x} \sim \{\mathcal{D}_n^k\}} \ell_{\text{NTP}}(\mathbf{x}; \boldsymbol{\theta}),$$

$$\mathcal{L}_{\text{OOD}}(\boldsymbol{\theta}) := \sum_{k \in [K]} \sum_{n \,:\, \mathbf{M}_{\text{in}}[n,k]=0} \mathbb{E}_{\mathbf{x} \sim \{\mathcal{D}_n^k\}} \ell_{\text{NTP}}(\mathbf{x}; \boldsymbol{\theta}).$$

During training, where we only get access to a subset of facts-template pairs $(k, n)$ for which $\mathbf{M}_{\text{in}}[n, k] = 1$, we are essentially minimizing $\mathcal{L}_{\text{ID}}(\boldsymbol{\theta})$. Intuitively this is the case because recall that we train the model such that at each iteration we see a fresh sequence $\mathbf{x}$ sampled from the ID templates and thus in the long run of many iterations, the training loss closely approximates the ID population loss $\mathcal{L}_{\text{ID}}(\boldsymbol{\theta})$. This is also empirically verified, since with sufficiently long training we always reach $100\%$ ID accuracies. Thus, during training we find model parameters $\boldsymbol{\theta}_{\text{ID}}$ that minimize the ID population risk, i.e.,

$$\boldsymbol{\theta}_{\text{ID}} \in \arg\min_{\boldsymbol{\theta}} \mathcal{L}_{\text{ID}}(\boldsymbol{\theta}).$$

We remark that the set of minimizers can possibly contain multiple solutions (and we shortly argue that it does!). Also note that in the assumed setting of $\mathcal{L}_{\text{tot}}(\boldsymbol{\theta}^*)$ attaining the total-loss lower bound, a minimizer $\boldsymbol{\theta}^*$ of $\mathcal{L}_{\text{tot}}$ is certainly a minimizer of the ID risk.

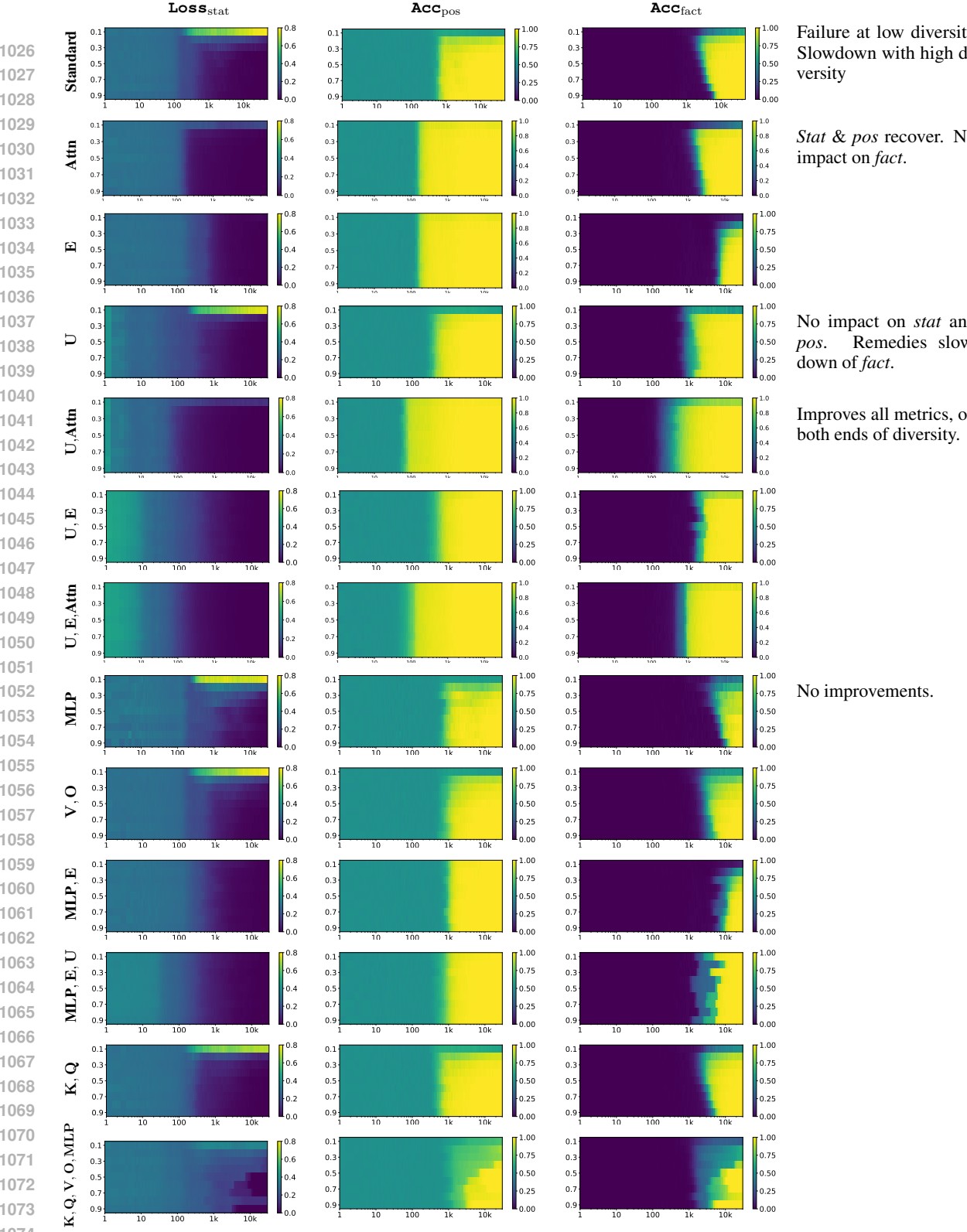

**Figure 9:** Experiments of Sec. 4 across different diversity levels and over training time. Each row label specifies which modules in the model, among the following, are frozen to that of $\theta_{\mathrm{hi}}$: softmax attention matrices **Attn**, token embedding / unembedding layer **E**/**U**, the weights of the fully-connected modules at each layer **MLP**, key, query, value, output matrices in self-attention module **K**, **Q**, **V**, **O**.

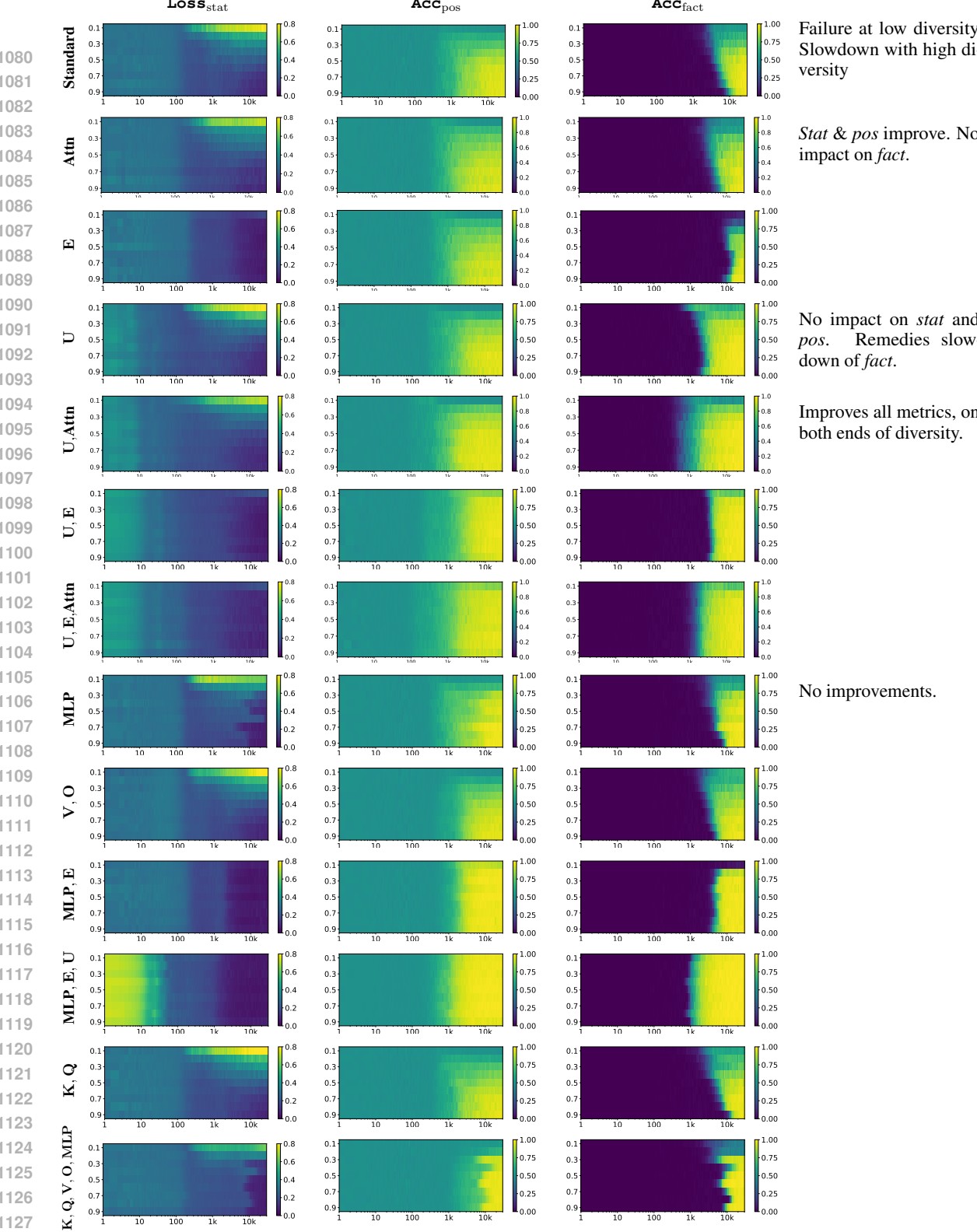

**Figure 10:** Same as the intervention analysis from Fig. 9 on a one-layer transformer. While the overall trends are similar to the four-layer model, the **Attn** intervention has a less significant impact here. This result aligns with the one-layer model's overall weaker performance (see also Fig. 7).

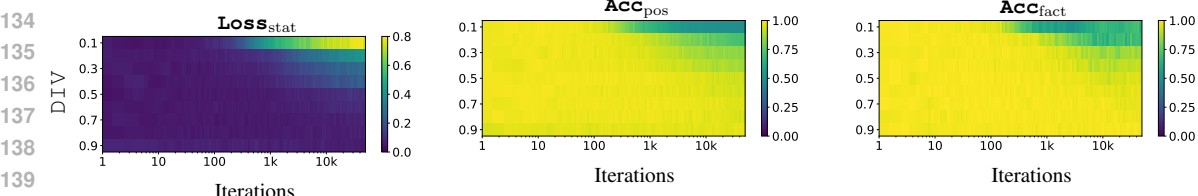

**Figure 11:** A one-layer transformer, initialized from a model trained on high-diversity data ($\texttt{DIV} = 0.9$), is then trained on a low-diversity dataset ($\texttt{DIV} = 0.1$). Although the model starts with perfect performance, its accuracy degrades across all metrics as training on the less diverse data progresses. This is similar to the degradation observed in Fig 6.

| | standard | MLP | U | Attn | E | E + U | Attn + U | Attn + E + U |
|---|---|---|---|---|---|---|---|---|
| $\mathrm{Loss}_{\mathrm{stat}}$ | 0.74 | 0.64 | 0.75 | 0.17 | 0.01 | 0.00 | 0.11 | 0.00 |
| $\mathrm{Acc}_{\mathrm{fact}}$ | 47.59 | 36.72 | 60.03 | 31.97 | 5.40 | 88.80 | 83.46 | 96.29 |
| $\mathrm{Acc}_{\mathrm{pos}}$ | 54.49 | 57.06 | 58.75 | 97.07 | 99.40 | 99.47 | 99.14 | 99.93 |

**Figure 12:** Same metrics as Fig. 5-(a), but captured earlier in training (10k vs. 30k iterations). This mid-training snapshot more clearly reveals that the intervention on **U** accelerates learning facts, even on low-diversity data that yields sub-optimal performance.

The interesting question is: *Does training find model parameters $\boldsymbol{\theta}_{\mathrm{ID}}$ that not only minimize the ID risk, but additionally minimize the OOD risk?* If that is the case, then $\boldsymbol{\theta}_{\mathrm{ID}} = \boldsymbol{\theta}^*$, i.e., $\boldsymbol{\theta}_{\mathrm{ID}}$ is a minimizer of the total loss $\mathcal{L}_{\mathrm{total}}$.

Our experiments (both in the original setup of Fig. 2 and even more evidently in the minimal setup of this section) reveal a compelling diversity-dependent dichotomy: On the one hand, under low-diversity, training converges to non-generalizing solutions that while they minimize $\mathcal{L}_{\mathrm{ID}}(\boldsymbol{\theta})$, they do not minimize the OOD risk $\mathcal{L}_{\mathrm{ID}}(\boldsymbol{\theta})$. Thus, $\boldsymbol{\theta}_{\mathrm{ID}}$ is a minimizer of the ID risk $\mathcal{L}_{\mathrm{ID}}(\boldsymbol{\theta})$ but a different one than the total loss minimizer $\boldsymbol{\theta}^*$. On the other hand, as diversity increases, training finds generalizing solutions, i.e. $\boldsymbol{\theta}_{\mathrm{ID}}$ is now a minimizer of both the ID and the total loss.

This dichotomy admits two possible explanations. With increasing diversity, either (1) the non-generalizing solutions are removed from the set of global optimizers of the ID loss $\mathcal{L}_{\mathrm{ID}}$, or (2) the landscape of the ID loss becomes more benign around the generalizing solutions (aka $\boldsymbol{\theta}^*$), which in turn makes it easier for the model to find them. Fig. 13-(b) also suggests that increased model capacity can partially help with making the ID landscape more benign.

Precisely characterizing how the context diversity and model capacity reshape the ID loss landscape is an exciting direction for future work. We believe the minimal setup and intuitions introduced in this section can serve as a starting point for such analysis.

# E    REPRESENTATION ANALYSIS

Building on our experiments probing the model's last-layer representations (Sec. 4.1), we now examine the clustering properties of hidden representations from all layers and the impact of training diversity.

To analyze the structure of these internal representations, for a given sequence drawn from the $n$-th template and carrying fact $(a_k, b_k)$, we probe hidden layer representations (for any layer $\ell$) at source fact position $i_n$ and test whether the transformer encodes a *template-invariant* representations of each fact $(a_k, b_k)$.

For every (fact, template) pair (both ID and OOD), we first sample $M$ sequences. For each sequence, we collect hidden vectors $\mathbf{h}_\ell^{(i_n)}$ at source fact position $i_n$ from the $\ell$-th transformer layer. For each layer $\ell$, we then stack the vectors into $\mathbf{H}^{(\ell)} \in \mathbb{R}^{P \times d}$ and keep the top $d' = \min(30, d, P)$ principal components, where $d$ denotes the dimensionality of the hidden layer representations, and

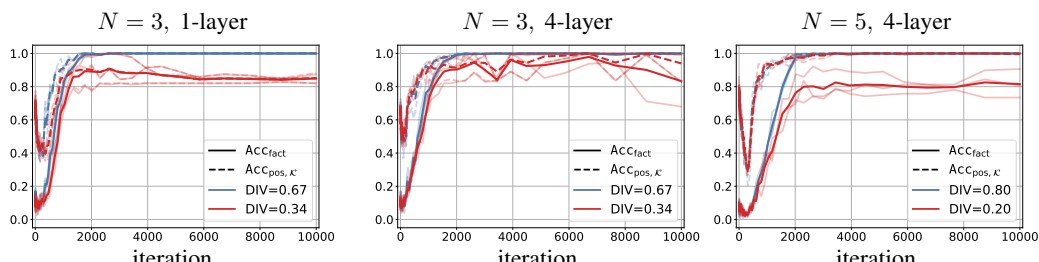

**Figure 13: Minimal setup to replicate the impact of diversity.** Factual recall $\mathbf{Acc}_{\text{fact}}$ (solid) and target-position accuracy $\mathbf{Acc}_{\text{pos},\mathcal{K}}$ (dashed). See Sec. D for discussion.

$P = M \times N \times K$ denotes the total number of hidden vectors extracted per layer. This gives us a PCA-reduced matrix $\widetilde{\mathbf{H}}^{(\ell)} \in \mathbb{R}^{P \times d'}$ where $\widetilde{\mathbf{h}}_i^{(\ell)}$, $i \in [P]$ denotes the PCA-reduced representation at fact position for the $i$-th sequence. Recall that $N$ denotes the number of templates, and $K$ denotes the number of atomic facts. We set $M$ to 250 in our experiments.

For every layer $\ell$, we take the PCA-reduced matrix $\widetilde{\mathbf{H}}^{(\ell)} \in \mathbb{R}^{P \times d'}$ and evaluate clustering quality of the vector embeddings hen they are labeled in two different ways: 1) each vector tagged with the fact index $k$, and 2) each vector tagged with the template index $n$. We measure the clustering quality with `silhouette_score` (SKLEARN). For a given hidden layer $\ell$, and every representation $\widetilde{\mathbf{h}}_i^{(\ell)}$, $i \in [P]$, we compute 1) $e_i^{(\ell)}$, the average Euclidean distance to all other vectors that share its label and 2) $f_i^{(\ell)}$, the smallest average distance to a group with a *different* label. Formally, if $C_i$ is the set of indices with the same label, then

$$e_i^{(\ell)} \;=\; \frac{1}{|C_i|-1} \sum_{j \in C_i, \, j \neq i} \left\| \widetilde{\mathbf{h}}_i^{(\ell)} - \widetilde{\mathbf{h}}_j^{(\ell)} \right\|_2, \quad f_i^{(\ell)} \;=\; \min_{C \neq C_i} \frac{1}{|C|} \sum_{j \in C} \left\| \widetilde{\mathbf{h}}_i^{(\ell)} - \widetilde{\mathbf{h}}_j^{(\ell)} \right\|_2.$$

Using these two metrics, the score for each vector embedding $\widetilde{\mathbf{h}}_i^{(\ell)}$ is defined as

$$s_i^{(\ell)} \;=\; \frac{f_i^{(\ell)} - e_i^{(\ell)}}{\max\{f_i^{(\ell)}, e_i^{(\ell)}\}} \in [-1, 1], \quad i \in [P].$$

The silhouette value attains 1 when $\widetilde{\mathbf{h}}_i^{(\ell)}$ lies well inside a compact cluster whose members share the same label, drops to 0 when clusters of different labels overlap, and becomes negative if the vector is closer to another label's cluster than to its own. The layer-level score $s^{(\ell)}$ is the average of these values across all vectors in the layer, i.e., $s^{(\ell)} = \frac{1}{P} \sum_{i \in [P]} s_i^{(\ell)}$. To differentiate the two labeling schemes, we denote the score as $s_{\text{fact}}^{(\ell)}$ when clusters are labeled using factual indices $k$, and as $s_{\text{tmpl}}^{(\ell)}$ when template indices $n$ are used as labels. If $s_{\text{fact}}^{(\ell)}$ is high, it indicates that the representations are *template-invariant*: the hidden representations learned for any given fact $a$ only depends on the fact itself and not the context template it appears in during training. In turn, if $s_{\text{tmpl}}^{(\ell)}$ is high, it suggests that the fact hidden representations from the same template cluster together even when the facts differ.

Figure 14 reveals three consistent trends in the clustering structure of hidden representations as training diversity grows. First, *factual identity is always the dominant organizing principle*: across layers, the fact curves sit well above the template curves, indicating stronger clustering by fact than by template. Second, this *fact-centric structure generalizes to unseen pairings*—the ID-only and ID+OOD fact curves roughly remain similar, showing that vectors from unseen fact–template combinations fall into the same clusters as their seen counterparts. Third, *greater template diversity progressively weakens template-based structure while fact-based structure remains intact*, so the gap between the two widens.

**Fact Heads.** To better understand how factual knowledge is stored across attention heads, we compute a per-fact head attribution heatmap. For each fact, we iteratively ablate individual heads (by zeroing their contribution) and measure the drop in the model's confidence for the correct token $b$. This is averaged over multiple in-distribution contexts where the fact appears, yielding a (fact × head) matrix

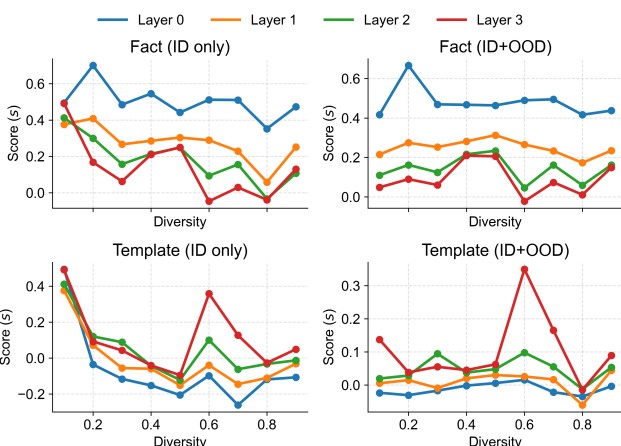

**Figure 14:** Clustering quality (see Sec. E) of hidden representations as a function of training diversity (DIV) for the MC1POS10 template type. **Top**: Clustering scores when representations are labelled based on factual identity $k$. **Bottom**: Clustering scores when representations are labeled based on template identity $n$. *Left* shows computation of score using representations of *ID* (template, fact) pairs. *Right* shows the clustering score using representations of both *ID* and *OOD* (template, fact) pairs. **Fact clustering dominates:** In both "Fact" panels (top row) every layer's curve sits well above the corresponding "Template" curves (bottom row). Hidden vectors therefore cluster primarily by the underlying fact rather than by the template. **Strong fact clustering persists on unseen templates:** The two fact curves—one computed on seen (ID-only) pairs, the other on the full ID + OOD set are roughly similar. Vectors for unseen fact–template combinations land in the same clusters as their seen counterparts, showing that the model abstracts the fact beyond the specific templates it saw during training. **Template invariance improves with diversity:** Moving from low to high diversity the template scores drift toward (or below) 0, while the fact scores remain roughly stable. This widening gap indicates that training on a broader mix of templates gradually removes template details from the representations while still keeping the facts separated.

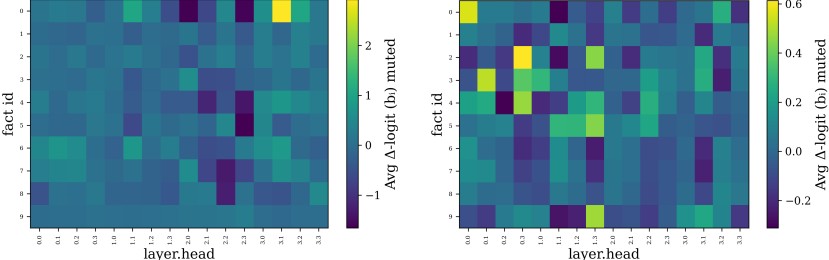

**Figure 15:** Per-fact head importance heatmaps in the factual recall task for the MC1POS10 template type. Each row corresponds to a different fact, and each column to a specific attention head (indexed as layer.head). The color indicates the average change in the logit of the correct answer token $b_i$ when the corresponding head is ablated, averaged over multiple in-distribution sequences for that fact. **Left**: model trained with low diversity (DIV = 0.1). **Right**: model trained with high diversity (DIV = 0.9). In the low diversity regime, head importance is diffuse and uniform across heads, suggesting no clear specialization. In contrast, at high diversity, certain heads become more consistently important for specific facts, indicating emergent specialization and more structured factual encoding.

of logit drops. Figure 15 shows these heatmaps for a model trained under low diversity (left) and high diversity (right). In the low diversity case, head importance is broadly distributed, with no head clearly emerging as critical for any fact. By contrast, in the high diversity regime, certain heads show strong and localized importance for specific facts, suggesting that the model has developed specialized storage heads. This points to a more structured encoding strategy that emerges only when the model sees the same fact across various different templates.

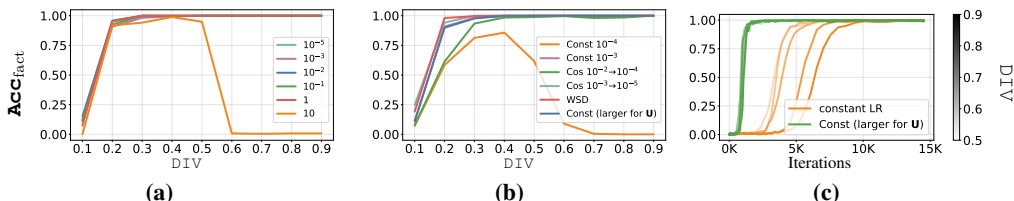

**Figure 16: Hyperparameter ablations.** OOD factual accuracy at the final training iteration (30K) with varying **(a)** weight-decay values and **(b)** learning rate values and scheduler. Performance remains largely unchanged across diversity levels. **(c)** Using larger learning rate for the unembedding layer **U** mitigates the training slowdown in higher diversity regimes. See App. F.1 for details.

# F ADDITIONAL EXPERIMENTS

## F.1 HYPERPARAMETER TUNINGS

**Hyperparameter changes do not resolve low-diversity failure**. In Figs. 16-(a,b), we examine whether standard optimization hyperparameters tuning can mitigate the failure modes observed in low-diversity regimes. We vary weight decay, learning rate, and learning rate schedulers, and find that none of these changes resolve the optimization bottleneck: when diversity is low, training consistently converges to a non-generalizing solution.

In Fig. 16-(a), we report OOD factual accuracy at the end of training (30K iterations) for weight decay values ranging from $10^{-5}$ to $10^{1}$. As long as weight decay is in a reasonable range, models trained on high-diversity data generalize on the OOD templates. In contrast, under low-diversity settings such as $\mathtt{DIV} = 0.1$ or $0.2$, factual accuracy remains uniformly poor and largely insensitive to the choice of weight decay.

Next, we observe the same pattern when varying the learning rate or the scheduler in Fig. 16-(b). We test four settings: (i) constant learning rates ($10^{-4}$, $10^{-3}$), (ii) a cosine schedule warming up from $10^{-4}$ to $10^{-3}$ and decaying to $10^{-5}$, (iii) a cosine schedule warming up from $10^{-3}$ to $10^{-2}$ and decaying to $10^{-4}$, and (iv) warmup–stable decay (WSD) (Hu et al., 2024) with a warmup from $10^{-4}$ to $10^{-3}$ followed by a stable phase and a final decay to $10^{-4}$. Across all these choices, performance remains effectively unchanged at each diversity level. The only exception is a constant rate of $10^{-4}$, where the training progress is slow and even high-diversity runs have not converged by the 30K-iteration mark.

**A larger unembedding learning rate accelerates training in high diversity regimes**. As discussed in Sec. 4, high-diversity training exhibits a slowdown: although models eventually achieve strong OOD generalization, they require more optimization steps. This slowdown disappears when the unembedding layer is fixed to the well-trained weights obtained from a high-diversity run, suggesting that the unembedding layer is a primary source of the optimization difficulty (Fig. 5).

Motivated by this observation, we train models across different diversity levels using a constant learning rate of $10^{-3}$ for all parameters, but assign a larger learning rate of $10^{-1}$ specifically to the unembedding layer. Fig. 16-(c) shows the resulting factual accuracy over training. Compared to the original training curves, the slowdown at high diversity is substantially reduced.

## F.2 VARIATIONS ON THE STATISTICAL STREAM

### F.2.1 HMM STATISTICAL STREAM

In this section, we replace the Markov-chain statistical stream with a hidden Markov model (HMM) to see the effect of having long-range dependencies in the statistical context. We focus on the MC10POS10 setup and for each template, we generate a distinct HMM with $H = 15$ hidden states and $V_{\mathcal{D}} = 10$ observable states. The hidden transition matrix and emission matrix are drawn independently from Dirichlet distributions with parameter $0.1$[1]. Aside from replacing the Markov chains with HMMs, all other aspects of the setup follow the experiment specification in Sec. 3.

---

[1]This choice yields relatively spiky transitions and emissions, producing more varied and context-dependent next-token distributions.

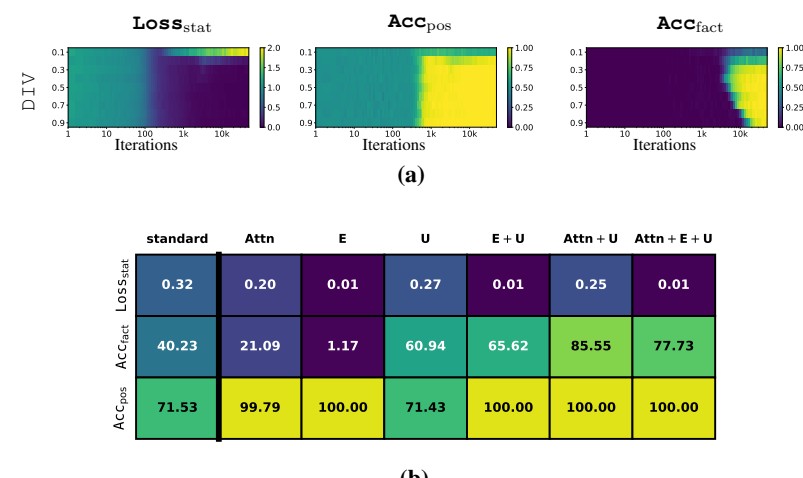

**(a)**

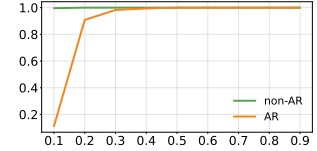

**(b)**

Figure 17: **Hidden Markov Models as statistical stream.** (a) $\mathbf{Loss}_{stat}$, $\mathbf{Acc}_{pos}$, and $\mathbf{Acc}_{fact}$. (b) Module intervention experiments analogous to Fig. 5: peak performance achieved during training. See App. F.2.1 for details.

Fig. 17-(a) shows $\mathbf{Loss}_{stat}$, $\mathbf{Acc}_{pos}$, and $\mathbf{Acc}_{fact}$ heatmaps across diversity and training iterations on OOD template-fact pairs. The heatmaps show that the qualitative effect of diversity remains unchanged: in low-diversity regimes, the model fails to generalize, while higher diversity improves OOD performance but results in a slower progress in performance, creating a temporal trade-off similar to the Markov chain case (Figs. 2 and 3).

The result of the targeted module interventions (following Sec. 4) in this setup also exhibits the same general pattern (Fig. 17-(b)): well-trained token embeddings and attention layers primarily improve positional and statistical generalization, while the unembedding layer is most directly associated with factual accuracy. We also find that reaching high factual accuracy with the module interventions is harder in the HMM setting: Interventions involving the unembedding layer improve factual accuracy considerably, but it does not lead to the full recovery seen in the Markov case (Fig. 5).

### F.2.2 REMOVING THE STATISTICAL COMPONENT

Here we argue that for studying the impact of diversity, it is necessary to consider the joint learning and modeling of the *statistical* and *factual* streams. Using the same data model as in Sec. 3, we modify the training objective so that the model is required to learn only the factual association. Concretely, we replace the statistical stream with random i.i.d. tokens drawn from the generic vocabulary, and we drop the autoregressive (AR) loss over the statistical tokens and compute the loss only at the position of the target fact token $b_k$. This setup is similar in nature to the setup considered in Nichani et al. (2024), with the main difference that factual tokens appear at predefined positions (i.e., templates) rather than at random locations.

Fig. 18 reports factual accuracy on sequences drawn from OOD template-fact pairs. In contrast to the AR setting, the need for diversity disappears: at the lowest diversity levels, the model achieves perfect factual accuracy on OOD templates. In other words, the sharp degradation in performance observed in low-diversity regimes throughout Sec. 3 does not occur when the model is trained only on the factual target.

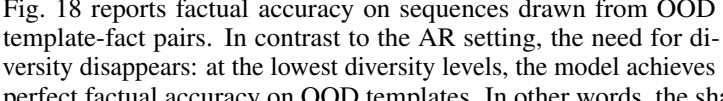

Figure 18: **Removing the statistical component** Factual accuracy at the final iteration for the non-AR objective (App. F.2.2) versus the standard autoregressive objective (MC10POS10 in Sec. 3). See App. F.2.2 for details.

### F.3 EFFECTS OF TOKEN AMBIGUITY

In the data model introduced in Sec. 2, the factual tokens and the generic statistical tokens come from disjoint vocabularies. However, in more realistic settings, a token might take different roles. To explore how such token ambiguity might affect the results, we modify the statistical stream so that a small fraction of factual tokens can also appear within it.

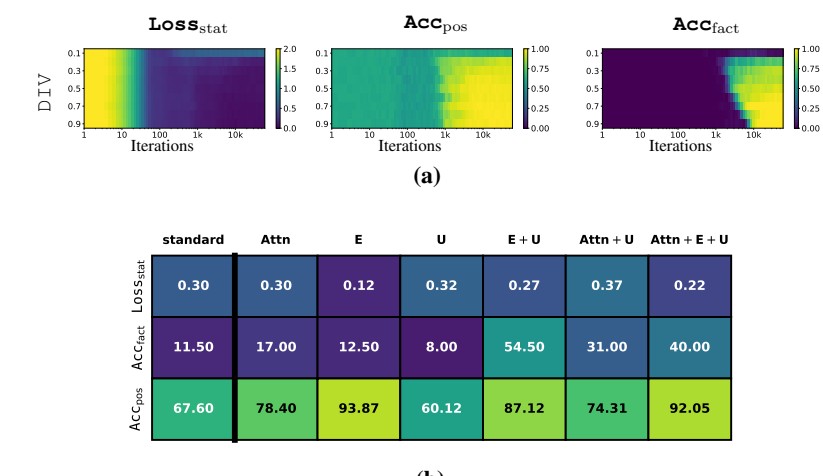

**(a)**

|  | standard | Attn | E | U | E + U | Attn + U | Attn + E + U |
|---|---|---|---|---|---|---|---|
| Loss$_{stat}$ | 0.30 | 0.30 | 0.12 | 0.32 | 0.27 | 0.37 | 0.22 |
| Acc$_{fact}$ | 11.50 | 17.00 | 12.50 | 8.00 | 54.50 | 31.00 | 40.00 |
| Acc$_{pos}$ | 67.60 | 78.40 | 93.87 | 60.12 | 87.12 | 74.31 | 92.05 |

**(b)**

**Figure 19: Impact of token ambiguity.** **(a)** Loss$_{stat}$, Acc$_{pos}$, and Acc$_{fact}$. **(b)** Module intervention experiments analogous to Fig. 5: peak performance achieved during training. See App. F.2.1 for details.

Concretely, for each experiment we select $20\%$ of the factual tokens (i.e., 40 tokens) uniformly at random and include them in the statistical vocabulary. The transition matrix of the statistical stream is then defined over the union of generic tokens and this overlapping fact subset. To limit the frequency with which these shared tokens appear as statistical predictions, we constrain the total transition probability mass to these tokens to be $10\%$. All other aspects of the experiment follow the setup of Sec. 3.

Fig. 19-(a) reports the OOD Loss$_{stat}$, Acc$_{pos}$, Acc$_{fact}$. The overall qualitative effect of diversity remains the same as in earlier experiments (Fig. 2): generalization fails in the low diversity regimes, and higher diversity improves OOD generalization at the cost of slower optimization. The key difference is that the model now requires *larger* diversity value to reach perfect performance across all metrics (see the less bright bands of factual accuracy for DIV $= 0.3 - 0.5$). The presence of token overlap increases the ambiguity between the factual and statistical streams, which appears to strengthen the reliance on diverse contextual exposure.

The intervention results in Fig. 19-(b) (analogous to Fig. 5) also follow the broad structure observed in previous sections. Token embeddings and attention contribute most strongly to positional and statistical improvement, and the unembedding layer remains most closely associated with factual accuracy. However, the magnitude of the improvements caused by these module-wise interventions is smaller compared to the disjoint-vocabulary setting.

