# OpenReview forum: "Facts in Stats: Impacts of Pretraining Diversity on Language Model Generalization"
_ICLR.cc/2026/Conference — Submitted to ICLR 2026_

### Official Review · Reviewer_Rbvz · 2025-10-31

**Soundness:** 3
**Presentation:** 3
**Contribution:** 2
**Rating:** 6
**Confidence:** 4

**Summary:**

This paper investigates how language models learn and distinguish factual knowledge from statistical patterns during pretraining. It investigates how the diversity of contexts in which a fact appears (i.e., paraphrasing) impacts both factual recall and statistical generalization. The paper's primary contribution is the introduction of a synthetic testbed that decouples the training data into a "statistical stream" (templates with controllable statistical) and a "factual stream" (atomic source-target pairs). This framework allows for fine-grained, independent control over the level of diversity (how many different templates a fact is seen in) and the contextual structure (how the templates themselves vary), which the authors argue was a key limitation in previous, less-controlled studies.

Using this testbed, the paper investigates the relationship between diversity and generalization. For example, while high diversity (seeing a fact in many different templates) slows down in-distribution (ID) factual learning, it is often crucial for out-of-distribution (OOD) generalization. A mechanistic analysis identifies that learned embeddings and the unembedding layer are the primary components enabling factual generalization under high diversity, while attention mechanisms are key for statistical learning. This work moves beyond prior synthetic studies, which often used fixed templates and focused solely on factual recall, by being the first to systematically model and analyze the interplay and trade-offs between statistical learning and factual acquisition.

**Strengths:**

The paper's primary strength is its experimental design. By cleanly decoupling statistical and factual streams, it offers fine-grained control over diversity and contextual structure — a significant advance over prior work. This design compellingly allows the authors to move beyond just factual recall and be one of the first to systematically investigate the interplay and trade-offs between statistical generalization and factual acquisition.
What's more, the paper provides a strong mechanistic analysis to explain why these trade-offs occur, identifying distinct optimization bottlenecks for each learning type . The fact that this minimal, reproducible testbed can still capture these phenomena (like stage-wise learning) seen in larger-scale studies is a plus.

**Weaknesses:**

My main concerns with this paper relate to its translation to realistic LLM training. My biggest question is about the distinct, non-overlapping vocabularies for statistical and factual tokens. This design removes the real-world ambiguity where a single token must participate in both roles, forcing the model to rely on context. [cite_start]I wonder if this simplification artificially affects the learning dynamics, perhaps exacerbating the observed importance of the (un)embedding layers in separating these tokens, rather than testing the internal processing required in a more realistic, mixed-vocabulary setting.

A secondary concern is the deliberately small model size. While I appreciate the authors' open acknowledgment and justification for this (tractability and reproducibility), and their checks on 1- and 10-layer models, it does leave open the question of how these specific bottlenecks might shift in the larger architectures.

**Questions:**

Do I understand correctly that the models are trained single-epoch? Whenever a statistical template is used, the actual tokens are sampled anew; and only the fact tokens are repeatedly found at their respective positions?

What is effective token frequency for fact vs. statistical tokens / how often do they appear throughout the whole training set? It appears with so many fact tokens and only few statistical tokes, there might be a significant imbalance?

---

> ### Author Response · Authors · 2025-11-21
>
> Thank you for your detailed feedback and positive review. We are glad that you found the synthetic experiment design, the strong mechanistic analysis, and the insights it provides useful.
>
> **Impact of overlapping vocab for fact and stat:**
> The non-overlapping design of our vocabularies was deliberate, as it provided the simplest possible setup to isolate the effect of contextual diversity from confounding factors like token ambiguity, thereby allowing for a cleaner evaluation of generalization aspects. This choice is also partly inspired by the observation in natural language that factual or highly specialized terms appear less frequently than generic ones, and when they do, their primary role is often to convey the specific fact. Nevertheless, we agree that in real corpora, there are tokens that might participate in both statistical and factual roles. As our claim/goal was not to fully model natural language, but to focus on the minimal aspects that are required for systematic study of diversity, we focus on the non-overlapping vocabularies.
>
>
> Motivated by your question, we have added new experiments (detailed in **App. F.3** and **Fig. 19**), where we allow for a 20% overlap between facts and generic tokens, enabling some tokens to appear in both the factual and statistical streams of the Markov chains. This requires refinements of our evaluation, as we would not have the separation of the metrics as clean as we had in the non-overlapping case. The key takeaways from this mixed-vocabulary experiment are that the core effects of diversity, namely, the generalization failure under low diversity and the temporal trade-off, remain qualitatively consistent. However, the model requires a higher level of diversity to achieve similar generalization performance. Furthermore, the fundamental mechanistic roles persist, e.g., the unembedding layer primarily improves factual accuracy, and attention/token embedding influences statistical generalization. However, the absolute effectiveness of such targeted interventions in improving generalization is reduced in this more complex setting.
>
>
> **Q) Is training a single epoch?**
> Yes, the models are effectively trained with a single epoch over the data, as the exact realization of any given sequence is rarely repeated. This is because the statistical tokens are freshly sampled at each training step. However, the model is nonetheless exposed to each in-distribution template-fact pairing multiple times.
>
> **Q) Effective token frequency and imbalance?**
> Yes, there is a frequency imbalance between the generic (statistical) tokens and the fact tokens in our current setup. Generic tokens appear roughly $\propto (T-2)/V_{\mathcal{D}}$, while fact tokens appear approximately $\propto 2/2K$. Given the typical values of $T$, $V_{\mathcal{D}}$, and $K$ in our experiments, the generic tokens are more frequent. This choice is also partly inspired by the observation in natural language that factual or highly specialized terms appear less frequently than generic ones.
>
> Thanks for your time to review the paper. We hope our responses address some of your concerns, and we appreciate your support.

---

### Official Review · Reviewer_CxgV · 2025-11-01

**Soundness:** 3
**Presentation:** 3
**Contribution:** 3
**Rating:** 6
**Confidence:** 4

**Summary:**

The paper studied the learning and generalization of "statistical structure" (different Markov chain structures) vs "facts" (particular tokens that ) in small transformers, using a synthetic setting designed to emulate those aspects of natural language pretraining data.. They swept across many parameters controlling data diversity (and also freezing various parts of the network after training in different regimes), and investigated the effects on learning and generalization of the "statistics" and "facts", and learning dynamics.

**Strengths:**

* very nice coverage of the parameter space. it can be difficult to study the effects of parameters on training in an exhaustive way, which they did well
* nice breakdown of the three types of learning: "positions" vs "statistics" vs "facts"
* interesting results, esp how diversity doesn't monolithically affect the different types of generalization

**Weaknesses:**

* limited scale
* Can you motivate more why it seems necessary to study the learning of facts and linguistic structure simultaneously? Why not e.g. facts vs reasoning, or facts vs learning larger scale statistical structure. Or each of them alone. Unclear why this is an important question
* I find the experimental setup very interesting. However, can you provide more precise motivation for why the Markov chain structure mirrors the statistical structure of natural language, while the two positions for "fact insertion" are a good representation of "facts"? Theoretically and/or empirically

**Questions:**

see above

---

> ### Author Response · Authors · 2025-11-21
>
> Thank you for your overall positive feedback. We are glad you found the results on the impact of diversity interesting and appreciated the capacity of our synthetic setup for breaking down the parameter space and different axes of generalization.
>
> We agree that many closely related questions are worth studying in their own right, and the examples you highlight are all important directions. Prior works have already begun to map this space. Recent works study how transformers learn pure statistical patterns, such as Markov chains, fixed (e.g., Makkuva et al. 2024) or in-context (e.g., Edelman et al. 2024; Park et al. 2025), as well as higher-order structures (e.g., Rajaraman et al. 2024), or different function classes, like linear regression and boolean functions (e.g., Garg et al., 2022). In parallel, other works focus on factual learning in isolation from the surrounding statistical structure, for example, by studying transformer memory capacity for atomic facts (Nichani et al. 2024), the dynamics of acquiring facts during pretraining (Zuccet et al. 2025).
> Our contribution in this work is adding another piece to this map: how data diversity shapes the learning dynamics or creates optimization challenges, motivated by the prior reports on the importance of data diversity for learning facts (Allen Zhu and Liu 2023).
>
> **Why does it seem necessary to study the learning of facts and linguistic structure simultaneously?** In realistic pretraining corpora, factual statements are not presented as isolated supervised examples. They are embedded in contexts whose structure is governed by linguistic or statistical regularities. The diversity of exposure to a given fact is therefore a property of the *contexts* in which the fact appears, not only of the fact itself. Thus, any attempt to study the impact of diversity on factual learning must account for some form of contextual statistics, which means that simultaneously modeling both facts and contextual structure is necessary.
>
> Beyond the necessity of incorporating both ingredients into the data design, we argue that it is also important to consider a setup where the model should *learn* them simultaneously. We support this with the new results in **App. F.2.2**, where we see that diversity no longer plays a critical role in learning factual associations if we modify the objective to only learn the fact tokens. Concretely, using the same data model of Sec. 3, we train the model non-autoregressively and compute the training loss only at the position of the target fact token $b_k$. In this setting. As shown in **Fig. 18**, even at the lowest diversity level, where each fact appears in only a single template during pretraining, the factual accuracy on sequences drawn from OOD template–fact pairs is close to perfect. The fact that the diversity effect vanishes in this isolated setting and reappears when the model must also learn the statistical stream shows that the phenomena we study need to take into account learning of the factual and statistical structure simultaneously.
>
>
> **Why the specific data design?**
> Following our discussion above, we do not mean for our data model design to be a full representation of language structure. Rather, we use natural language as an inspiration to derive the *simplest* setup we found that reproduces the necessity of contextual diversity for *factual* generalization (i.e., predefined associations between concepts/tokens, a simplified form of knowledge graphs). Here, we see Markov chains to be a simple enough statistical stream that allows us to capture this phenomenon and provides a fully controllable testbed for the systematic dynamics and mechanistic analysis that we present. This simplicity is precisely what allows us to study the parameter space systematically and to cleanly separate positions, statistics, and facts, which you highlighted as the strengths of our work.
>
> Thanks for your effort to review the paper. We hope our responses address some of your concerns, and we appreciate your support.

---

> > ### Comment · Reviewer_CxgV · 2025-11-24
> >
> > Thank you for your responses. I appreciate your answer re learning facts and structure simultaneously.
> >
> > However, I am still unconvinced that the setup is a good representation of the phenomena that they are supposedly reprentative of -- I appreciate the desire to have simplicity and controllability, but you still need to be reasonably modeling the phenomena that you are making claims about. This needs to be better motivated, or the claims need to be adjusted.
> >
> > I will maintain my score.

---

> > > ### Author Response · Authors · 2025-11-25
> > >
> > > Thank you for the response and for the overall positive score, which we appreciate.
> > >
> > > To reiterate, the phenomenon we aim to study is the impact of pretraining diversity on generalization. Our synthetic setup is designed to represent and replicate the two key ingredients that, based on prior work, are most relevant to this question: (i) the importance of contextual diversity in learning predefined factual associations between tokens, and (ii) the basic learning dynamics of such factual associations (e.g., Allen-Zhu and Li, 2023; Nichani et al., 2024; Zucchet et al., 2025).
> > >
> > > Throughout the paper and in the rebuttal, we have aimed to carefully contextualize and justify our claims within this setup, and we will make this more explicit. If you have specific suggestions, we welcome your valuable input.
> > >
> > > Thanks again.

---

### Official Review · Reviewer_q86T · 2025-11-01

**Soundness:** 3
**Presentation:** 2
**Contribution:** 2
**Rating:** 4
**Confidence:** 3

**Summary:**

The author investigate how linguistic diversity influences factual and statistical learning in language models. They design a controllable synthetic testbed separating a statistical stream and a factual stream, allowing independent control of context structure and diversity level. They that higher diversity delays factual recall within distribution but is necessary for robust out-of-distribution generalization, while low diversity can cause failures in both factual and statistical learning.

**Strengths:**

- To study this, the authors design a controllable synthetic framework that separates statistical structures from factual associations, enabling the study of their interaction during training.
- They systematically examine how diversity level and contextual structure affect in-distribution.

**Weaknesses:**

- I think the setting is a bit overly simplified, Markov templates and atomic facts miss real linguistic hierarchy and semantic interference---this could limit the external validity.
- The uniform sampling of diversity overlooks long-tailed frequency patterns in real corpora, this could possibly misrepresent the true diversity effects.

**Questions:**

What diversity level is actually needed to improve generalization? Does the model learns true fact template separation or just memorizes within templates?

---

> ### Author Response · Authors · 2025-11-21
>
> Thank you for taking the time to review our paper. We appreciate your recognition of the value of a controllable synthetic framework and the systematic analysis it enables. We respond to your comments below.
>
> **Overly simplified fact and stat model:**
> We acknowledge the simplicity of the setup, but this simplicity is intentional:
> Our goal in this work was to isolate and study the impact of pretraining diversity on a transformer’s ability to learn facts systematically. For this purpose, our aim was to design the *simplest* setting that preserves the two elements needed for this question: (i) the importance of contextual diversity in learning facts, as highlighted by Allen-Zhu and Li (2023), and (ii) the basic learning dynamics of factual associations observed in prior work, such as Zuccet et al. (2025).
> We do not aim or claim to fully model natural language; rather, we take inspiration from its structure and retain only the essential properties necessary for the phenomenon we want to study.
>
> In this sense, we see the simplicity as a feature: it reproduces the essential ingredients of the problem, isolates the effect of diversity from other confounding factors, and provides a controlled environment where different axes of generalization can be examined clearly, which is a strength also noted in your review. It is this tractability that allows us to do an exhaustive analysis in this setup and study how changes in data structure, vocabulary sizes, model sizes, the interaction between streams (Sec 3) and different model components affect training dynamics (Sec. 4). We have also added experiments with HMM statistical streams (**Fig. 17 in App. F.2.1**), where the next-token probabilities are less trivial than markov chains and depend on the full context history while remaining tractable. These experiments show that our main observations are not specific to the simplest statistical structures.
>
> Overall, we view this framework as a starting point for systematic investigation rather than an endpoint. Some insights will carry over to more complex settings, while others may not. However, we expect this type of analysis to be an iterative process: first identifying important phenomena in complex, large-scale settings, and then isolating the essential components to reproduce these behaviors in the simplest, yet tractable, setup. This simplified setup facilitates finer-grained studies and discoveries, which are inherently valuable as they directly illuminate the fundamentals of transformers and AR training.
>
> **Uniform sampling of diversity:**
> Real corpora indeed exhibit heavily imbalanced/long-tailed properties. However, imbalances typically make training and generalization more difficult, not easier. A configuration that fails to learn or generalize (e.g., here to new fact-template pairs) under a balanced regime is therefore likely to fail, often more severely, under a more realistic imbalanced setup.
>
> Our central goal in this paper is to isolate the specific impact of diversity on the training dynamics of transformers. To do this rigorously, it is necessary to control for frequency as a confounding variable. With an imbalanced distribution, it would be difficult to pinpoint the exact nature of failure – lack of diversity or the imbalanced properties of the pretraining distribution. That’s why we choose the frequencies of individual facts and their diversity levels to be uniform across templates.
>
> We agree that introducing long-tailed properties is a promising direction for better modeling of real corpora. However, even within this balanced regime, we found the behavior of the models under different diversity settings to be rich, meriting a dedicated study on its own. As briefly noted in the conclusion section, we find our framework well-suited for a fine-grained investigation of other factors impacting generalization, like imbalances, misrepresentations of facts, and fine-tuning on new templates. For instance, for the imbalanced setup, one can simply modify the exposure mask to incorporate frequency imbalances, and we view this as a natural and important next step for future work.

---

> > ### Author Response · Authors · 2025-11-21
> >
> > **Q) “What diversity level is needed?”** This is exactly the question we study: how generalization changes *qualitatively* as we move along the diversity spectrum. If the intent is to identify a single *quantitative* threshold, that depends on the task, the definition of diversity in that task, and the training regime. Even in our controlled setting, where diversity has a clean definition, the diversity level that yields the best performance varies with several factors, as pointed out in the paper: Higher diversity provides a better representation of the full underlying distribution, but its optimality is influenced by practical constraints. For example, under a low training budget, high diversity can be suboptimal (Sec. 3.2), and changing the model size shifts the range of diversity levels that achieve strong performance within a fixed budget (Fig. 7).
> >
> > **Q) “Does the model learn true template-fact separation?”** The distinction between memorization and true fact–template separation is already reflected in our separate ID and OOD evaluations. A model that only memorizes template–fact pairs cannot perform well on OOD templates, since those combinations never appear during training. The improvement in OOD performance as diversity increases, therefore, indicates that the model begins to learn the separation of the underlying streams.
> >
> > We hope that these responses address your concerns and that you will consider increasing your scores.

---

> > > ### Author Response · Authors · 2025-11-27
> > >
> > > Dear Reviewer q86T,
> > >
> > > Thank you again for your time.
> > >
> > > We hope our responses have addressed your concerns, and we hope that you reconsider your evaluation. We appreciate any additional feedback, and we are happy to clarify anything that may still be unclear.

---

### Official Review · Reviewer_oCE1 · 2025-11-04

**Soundness:** 3
**Presentation:** 2
**Contribution:** 2
**Rating:** 4
**Confidence:** 3

**Summary:**

This paper investigates the impact of pretraining data diversity on language model generalization. The authors introduce a novel synthetic testbed that combines statistical streams (templates) with factual streams (source target pairs). This setup allows fine grained control over the diversity level, meaning how many different templates a fact appears in , and the contextual structure, meaning how templates vary statistically or positionally. Key findings show that while high diversity can slow in distribution fact learning , it is critical for out of distribution generalization. The specific effects depend heavily on the contextual structure.

**Strengths:**

1. The synthetic testbed is a primary strength. It is cleverly designed to isolate and control the interactions between statistical learning and factual memorization.
2. The paper provides a nuanced analysis of diversity. The finding that high diversity slows ID convergence but is essential for OOD generalization is a valuable insight.
3. The discovery of a temporal trade off where optimal diversity levels depend on the training duration is an interesting and practical finding.

**Weaknesses:**

1. The primary limitation is the simplicity of the synthetic data. While tractability is a goal , the first order Markov chain used for the statistical stream  is a very simplified model of language. It is unclear how these findings translate to the complex, long range dependencies of natural text.
2. The experiments are conducted on very small transformer models. Although the authors show results on 1 to 10 layer models , it remains an open question if these specific mechanisms and bottlenecks are the same in SotA models with hundreds of billions of parameters.
3. The paper's findings are dense. While the figures are informative, the interplay between the three contextual structures (MC10Pos1, MC1Pos10, MC10Pos10) and the three metrics  can be difficult to follow.

**Questions:**

1. The paper convincingly argues that low diversity creates a non generalizing minimizer . Could this be overcome with optimization changes, such as a different optimizer, learning rate schedule, or regularization, or is it a fundamental property of the low diversity data landscape?
2. How do you hypothesize these findings would change if the statistical stream was more complex than a first order Markov chain? For example, if it included simple grammatical structures, would the sharp distinction between statistical and factual learning bottlenecks  still hold?

---

> ### Author Response · Authors · 2025-11-21
>
> Thank you for your time and detailed feedback. We are pleased that you found our synthetic testbed and the resulting fine-grained analysis and discoveries to be interesting. We have made several changes to the paper and added new experiments in App. F, inspired by your questions, which we hope address your concerns. We respond to each of your comments separately below.
>
> **(W1-2 and Q2) simplicity of data and scale of model:** The simplicity of data design and model adopted in our work was a deliberate design choice: as also acknowledged in your review, our primary goal was to isolate and systematically study the impact of pretraining diversity on a transformer's ability to acquire facts. To achieve this, we aimed for the *simplest* possible setting that still preserves the essential phenomena reported in prior studies, namely, the importance of contextual diversity for factual learning and its core dynamics. In this sense, simplicity is a feature that allows us to fully control and isolate the effect of diversity from confounding factors, and enables the fine-grained mechanistic analysis presented in the paper.
>
> Prompted by your comment on the Markov Chain (MC)'s lack of long-range dependencies, we have added new experiments using Hidden Markov Model (HMM) as statistical streams **(Fig.17)**. This modification makes the next-token probabilities less trivial, as they depend on the full context history, yet keeps them tractable.
>
> The results demonstrate that our main observations are not specific to the simple MC model. Specifically, the impact of diversity is similar: in low-diversity regimes, the model fails to generalize, while higher diversity improves OOD performance but results in slower progress in performance, creating a temporal trade-off similar to the MC case. However, the increased complexity of the statistical stream makes achieving perfect factual accuracy slower.
>
> The impact of the targeted module interventions also largely remains consistent: in particular, transferring attention or token embeddings primarily enhances the statistical and position metrics, while transferring unembedding weights specifically improves the factual accuracy, but achieving perfect factual accuracy is harder in this setup. We provide more details on this experiment in **App. F.2.1**.
>
> Thank you for the suggestion. We agree it is important to test the robustness of these findings at various levels. The HMM experiment serves that purpose, and future work could investigate such specifics in large-scale real data settings. Overall, we expect this type of analysis to be an iterative process: first identifying important phenomena in complex large-scale settings, and then isolating the essential components to reproduce these behaviors in the simplest tractable setup. This simplified setup facilitates finer-grained studies and discoveries, which are inherently valuable as they directly illuminate the fundamentals of transformers and AR training.
>
> **(W3) findings are dense:**
> We appreciate your feedback. To enhance the clarity and flow of the presentation, we have updated Fig. 1 to more cleanly discuss the roadmap of the sections and provide a concise overview of the setup and key results. We have also expanded the summary Sec. 1.1 to provide a more detailed breakdown of all the results in the paper. This being said, we would be happy to discuss any specific concerns.
>
> **(Q1) Impact of hyperparameters on low-diversity failure:**
> Thanks for the question. By the time of the initial submission, we had tested some standard adjustments on the optimization parameters, including different learning rates and weight decay. We did not initially include them in the paper, as none proved effective in low-diversity regimes for finding generalizing minimizers. However, prompted by your question, we complement our initial observations with new experiments, which are presented in **Fig. 16-App.F.1** and briefly summarized below:
>
> In the low-diversity regime, tuning the value of weight-decay and learning rate, or choosing different schedulers (cosine decay, warm-up stable decay), does not have a significant impact on OOD generalization.
>
> Interestingly, our investigation into optimization dynamics revealed a separate, relevant insight about the high-diversity regime. Inspired by the Section 4 experiments, where we saw that with unembedding interventions, the models no longer show slowed learning with increasing diversity levels, we retrain the models from scratch using a distinctly larger learning rate applied only to the unembedding layer. This learning rate adjustment shows a similar effect, and the speed of factual learning becomes largely robust to the diversity level. It is important to note, however, that the original low-diversity failure, the inability to find a generalizing minimizer, persisted even with this modification.
>
> We hope that these responses address your concerns and that you will consider increasing your scores.

---

> > ### Author Response · Authors · 2025-11-27
> >
> > Dear Reviewer oCE1,
> >
> > Thank you again for your time.
> >
> > We hope our responses have addressed your concerns and we hope that you could reconsider your evaluation. We appreciate any additional feedback, and we are happy to clarify anything that may still be unclear.

---

### Author Response · Authors · 2025-11-21
**Updates to the Paper**

We thank all the reviewers for their time and feedback. We have revised the paper and added new experimental results in the Appendix to address several concerns raised in the reviews. The major updates and new material are listed below, with new sections highlighted in blue in the PDF:

- **Updated Introduction**: We have updated Figure 1 and the contribution summary in Section 1.1 to provide a clearer roadmap and a more detailed overview of the paper's key results.

- **Impact of Hyperparameter Tuning (Appendix F.1):** We include new experiments investigating the impact of hyperparameter tuning, considering both the high-diversity and low-diversity regimes.

- **Variations on the statistical stream (Appendix F.2):** We add new experiments replacing the first-order Markov Chain (MC) with Hidden Markov Models (HMM) to study the impact of long-range dependencies in the statistical stream.

- **Token Ambiguity (Appendix F.3):** We provide new experiments with token ambiguity, where the generic (statistical) and fact vocabulary are not disjoint and share an overlap.

---

### Meta-Review · Area_Chair_r2V4 · 2026-01-14

**Summary:**

The reviewers are relatively mixed overall with no strong opinions for or against the paper. They agree that the testbed is a primary strength and the overall comprehensiveness is great. But they feel that the simplicity and small scale of the experiments. It is also not clear how well the setup approximates more realistic data.

Overall their concerns included:
* The simplicity of the synthetic data.
* The experiments are conducted on very small transformer models.
* The paper's findings are dense. The interplay between the three contextual structures (MC10Pos1, MC1Pos10, MC10Pos10) and the three metrics can be difficult to follow.
* The uniform sampling of diversity overlooks long-tailed frequency patterns in real corpora
* The distinct, non-overlapping vocabularies for statistical and factual tokens. This design removes the real-world ambiguity where a single token must participate in both roles

**Reviewer Concerns:**

The majority of the concerns above remain open, although the authors did answer many other questions the reviewers posed convincingly.

**Reviewer Scores:**

I do not think the scores would change significantly in either direction.

---

### Decision · Program_Chairs · 2026-01-26

Reject